# Structural dynamics of DNA strand break sensing by PARP-1 at a single-molecule level

Anna Sefer[1,5], Eleni Kallis[1,5], Tobias Eilert[1,2], Carlheinz Röcker[1], Olga Kolesnikova[3], David Neuhaus [4], Sebastian Eustermann [3] ✉ & Jens Michaelis [1] ✉

Single-stranded breaks (SSBs) are the most frequent DNA lesions threatening genomic integrity. A highly kinked DNA structure in complex with human PARP-1 domains led to the proposal that SSB sensing in Eukaryotes relies on dynamics of both the broken DNA double helix and PARP-1's multi-domain organization. Here, we directly probe this process at the single-molecule level. Quantitative smFRET and structural ensemble calculations reveal how PARP-1's N-terminal zinc fingers convert DNA SSBs from a largely unperturbed conformation, via an intermediate state into the highly kinked DNA conformation. Our data suggest an induced fit mechanism via a multi-domain assembly cascade that drives SSB sensing and stimulates an interplay with the scaffold protein XRCC1 orchestrating subsequent DNA repair events. Interestingly, a clinically used PARP-1 inhibitor Niraparib shifts the equilibrium towards the unkinked DNA conformation, whereas the inhibitor EB47 stabilizes the kinked state.

Accurate sensing of DNA lesions is essential for genome maintenance across all kingdoms of life. High-resolution structures of DNA lesions in complex with sensor proteins have provided key insights into the underlying mechanism of damage recognition. Notably, many of these structures exhibit highly perturbed, non-B-form DNA conformations in the bound state that differ markedly from their respective unbound DNA states. A prominent example is the binding of a G·T mismatch by the mismatch repair protein MutS, which causes the damaged DNA to adopt a kinked state[1]. Similarly, a nicked DNA bound by the Flap endonuclease Fen1 is sharply kinked at the damage site[2]. DNA kinking is also observed upon Rad4/XPC binding to damaged DNA[3]. Thus, altered DNA deformability often plays a major role for DNA damage recognition.

In order to unravel the role of such perturbations in DNA damage recognition, single-molecule studies have been employed. Single-molecule Förster resonance energy transfer (smFRET) data has provided further insight into DNA kinking upon binding of MutS protein, showing that the DNA fluctuates rapidly between different bent states[4,5]. DNA conformational dynamics are also important during Fen1

binding to a DNA flap, which results in a change in DNA kinking that in turn leads to flap junction opening[6–8]. Remarkably, conformational changes of DNA during DNA damage recognition are also observed in the context of chromatin, since the UV-damaged DNA-binding (UV-DDB) protein complex accesses inward lesions on the nucleosome by shifting the nucleosomal DNA[9]. All these examples show that perturbations in DNA conformation often play a key role in DNA damage recognition. However, even in light of this progress as yet limited mechanistic insight was obtained about the pathway by which the protein-DNA complex reaches this state, leaving key determinants of DNA damage recognition elusive.

Understanding the mechanistic basis of DNA damage sensing is particularly relevant for Poly(ADP-ribose)Polymerase-1 (PARP-1). PARP-1 is a highly abundant and conserved nuclear stress response protein that constitutes the principal sensor of DNA single-stranded breaks and other types of DNA lesions in higher Eukaryotes. Moreover, it is also involved in a plethora of other nuclear processes, including transcription regulation as well as chromatin organisation[10–14]. Upon binding to DNA damage, or by other signals, PARP-1 becomes

[1]Institute of Biophysics, Ulm University, Albert-Einstein-Allee 11, 89081 Ulm, Germany. [2]Boehringer Ingelheim, CoC CMC Statistics & Data Science, Birkendorfer Str. 65, 88400 Biberach, Germany. [3]European Molecular Biology Laboratory (EMBL), Heidelberg Meyerhofstraße 1, 69117 Heidelberg, Germany. [4]MRC Laboratory of Molecular Biology, Francis Crick Avenue, Cambridge Biomedical Campus, Cambridge CB2 0QH, UK. [5]These authors contributed equally: Anna Sefer, Eleni Kallis. ✉e-mail: jens.michaelis@uni-ulm.de; sebastian.eustermann@embl.de

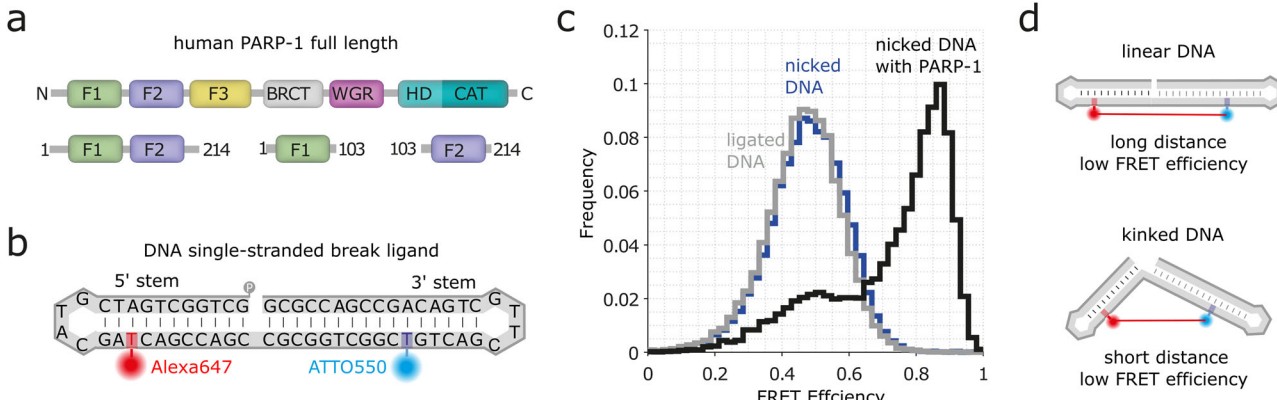

**Fig. 1 | smFRET assay for probing the DNA conformation in presence of PARP-1.** **a** Domain architecture of full-length PARP-1 and PARP-1 fragments F2 and F1F2. Domains are abbreviated as: F1, F2, F3: zinc finger domains; BRCT: breast cancer susceptibility protein C-terminal domain; WGR: WGR domain; HD: α-helical subdomain; ART: ADP-ribosyl transferase subdomain. **b** Schematic displaying the sequence and predicted secondary structure of the DNA ligand used in this study. The DNA is labeled with Alexa647 (red) at position T18 and with ATTO 550 (blue) at position T37; both dyes are attached via a C2 linker (Methods). **c** smFRET efficiency histogram obtained from freely diffusing molecules of the nicked DNA (blue), DNA after highly efficient ligation of the nick (grey, Supplementary Fig. 1), and of the nicked DNA in the presence of full-length PARP-1 (black). **d** Schematic depicting how changes in DNA conformation can be detected by smFRET.

allosterically activated and catalyzes a post-translational modification, in which poly(ADP-ribose) (PAR) is synthesized and attached to sidechains in proteins (including PARP-1 itself), thus directly modulating protein function and leading to the recruitment of various downstream factors[15]. Inhibition of PARP-1 is synthetically lethal for cells deficient in DNA repair factors (e.g. BRCA1) and other stress response proteins. Such deficiencies are frequently caused by cancer mutations. Therefore, PARP-1 inhibitors are successfully used in cancer therapy and serve as a paradigm for the development of a novel generation of drugs exploiting the principle of synthetic lethality[16–19].

The dynamic multi-domain architecture of PARP-1 is thought to play a central role for DNA single-strand break (SSB) recognition, being crucial for PARP-1 allosteric activation, for its concerted interplay with other DNA damage response and repair factors, as well as for the design and function of PARP-1 inhibitors. PARP-1 consists of six domains (Fig. 1a), namely two N-terminal zinc fingers (F1 and F2), a third (structurally unrelated) zinc finger (F3), a BRCT-domain (breast cancer susceptibility protein C-terminal domain) containing automodification sites, a WGR-domain, and the C-terminal catalytic domain. The catalytic domain comprises a regulatory helical subdomain (HD) and the ADP-ribosyl transferase subdomain (ART). When free in solution, these six domains are largely non-interacting, behaving like "beads-on-a-string"[20], but once F1 and F2 recognize a DNA lesion, all domains, apart from the BRCT domain, assemble into a welldefined structure at the damage site by forming interdomain contacts[21–24]. This multi-domain assembly cascade acts as a allosteric switch[23] which releases auto-inhibition of the C-terminal catalytic domain through a conformational change of its HD subdomain allowing PARP-1 to reach its catalytically active state[21,24–27]. The importance of PARP-1's dynamic nature is also underlined by the fact that the lethal effect of PARP-1 inhibitors is thought not to be primarily due to a lack of PAR signaling but rather to the ability of the inhibitor to trap the enzyme at the DNA damage site by preventing PAR synthesis, and/or strengthening the protein-DNA binding[18,28].

The structure of F1F2 bound to a 1nt gap DNA[23] currently remains the only high-resolution structure of any PARP-1 domain in complex with an SSB, although other forms of SSBs are also known to activate PARP-1, e.g. nicks or longer gaps[29,30]. In this structure, the gapped DNA adopts a highly kinked conformation, similar to what had been seen for nicked DNA in early electron microscopy studies[31]. F2 is bound to the side of the gap bearing the 3′ terminus, F1 binds to the side bearing the 5′ terminus, and the two zinc fingers form interdomain contacts with

one another. Notably, the two flexibly linked zinc finger domains show no interdomain interaction in the absence of DNA. Consequently, kinking of the DNA cannot be explained by binding to a relatively rigid, pre-formed protein interface, as proposed for other DNA damage sensor proteins (see above). Instead, it points towards a more elaborate mechanism in which altering the dynamics of the damaged DNA as well as the multi-domain organization of PARP-1 is important for DNA damage recognition. It has been speculated that F2 is needed for the first step of SSB recognition, involving the opening of the DNA around the SSB to make the F1 binding site accessible[23,32].

Broadly similar considerations apply to understanding the mechanisms of action of inhibitors, since it has been shown that these differ not only in their ability to prevent PAR synthesis but also, importantly, in how they modulate the DNA-dependent domain assembly process underlying activation. Zandarashvili et al.[18] have defined three classes of PARPi according to the allosteric effect they exert on the strength of the PARP-1 binding to DNA damage: class I, also called "pro-retention", which increases the affinity of PARP-1 binding to DNA damage; class II or "neutral" which leaves it predominantly unchanged, and class III, also called "pro-release" which weakens it.

During SSB repair, the X-ray repair cross-complementing protein 1 (XRCC1) is recruited to the SSB in a PARP-1 dependent manner. This recruitment is primarily driven by an interaction between XRCC1 and newly formed PAR chains[33,34], but may involve also direct XRCC-1 binding to damaged DNA[35,36]. XRCC1 acts as a scaffold protein which orchestrates, through an unknown mechanism, a "hand-off" to other core factors involved in repair of SSBs, such as DNA polymerase β and DNA ligase IIIα[11], while it plays also a key role in preventing PARP-1 from being trapped at the site of DNA lesions[37].

Many central questions regarding the mechanism of DNA damage recognition by PARP-1 remain under discussion: does the mechanism of DNA damage recognition resemble a conformational selection or rather an induced fit, how does the behavior of the dynamic multidomain structure contribute to this process, and do PARP inhibitors modulate DNA damage recognition through allosteric effects? In this study, we combined smFRET experiments, structural modeling and computational approaches to gain direct insight into the binding of PARP-1 to a nicked DNA. Our data show that in the absence of protein, the DNA molecule adopts a conformational ensemble almost identical to linear undamaged DNA. We further show that when the F2 domain alone binds, it kinks the DNA, while binding of F1F2 kinks the DNA still

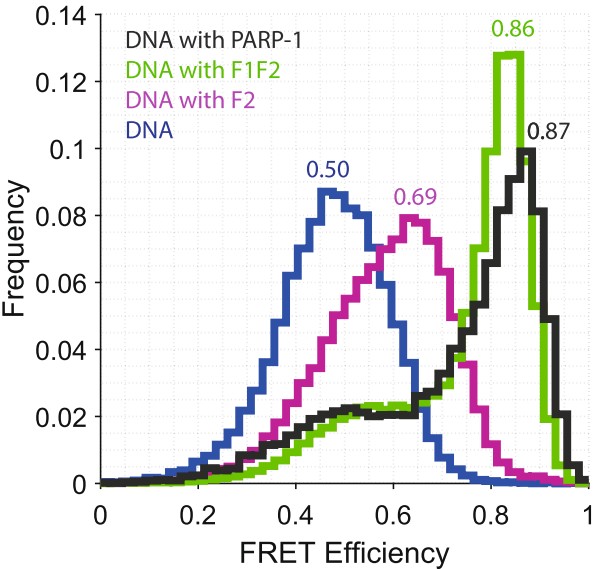

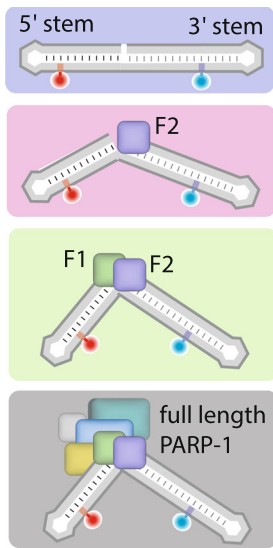

**Fig. 2 | DNA conformation during nick recognition probed by smFRET.** smFRET efficiency histogram of nicked DNA (blue) and nicked DNA in the presence of either F2 (magenta), F1F2 (green) or full-length PARP-1 (grey here, the same histogram as in Fig. 1), together with cartoons illustrating the underlying conformations. The indicated values correspond to the peak FRET efficiencies obtained by Gaussian fitting (Methods, Supplementary Fig. 2 and Supplementary Table 1).

further. By implementing a hybrid smFRET and computational approach, we determined the kinking angle of the DNA in the contexts of these different complexes. Time-resolved single molecule fluorescence also sheds light on the dynamics in these complexes and provides direct evidence that PARP-1 binding does not involve conformational selection, but instead rather resembles an induced fit mechanism. Furthermore, the functional importance of PARP-1 dynamics is supported by smFRET experiments in the presence of PARP-1 inhibitors showing distinct dynamics for different classes of clinically used inhibitors. We also show that XRCC1 binding to DNA lesions is stimulated by F1F2 and that XRCC1 binding in presence of F2 changes the observed distribution of DNA states which may contribute to a "hand-off" to other DNA repair factors.

## Results

To dissect DNA SSB recognition by PARP-1 at the single-molecule level, we used a single-stranded DNA molecule designed to form a dumbbell structure, in which an SSB is created between two hairpins (in the following, the stem carrying the free 5′ terminus is called the 5′ stem and that carrying the free 3′ terminus is called the 3′ stem) (Fig. 1b, Methods). The positions of two fluorophores were optimized on either side of the nick (18 bases apart), ATTO 550 on the 3′ stem and Alexa647 on the 5′ stem, respectively, to sensitively monitor DNA conformations by measuring the smFRET efficiency. Using time-resolved fluorescence spectroscopy, smFRET efficiencies were determined from free DNA as well as from the DNA in the presence of saturation concentrations of PARP-1 and fragments thereof (Fig. 1c, Methods). Due to the design of the DNA ligand, smFRET data can be used to assess the kinking angle between the two DNA stems (Fig. 1d).

Comparison of the smFRET data obtained from the DNA containing the SSB to that of a ds-DNA molecule obtained by ligation with T4 DNA ligase yielded virtually identical smFRET efficiency histograms (Fig. 1c), with peak FRET efficiencies of $E = 0.50 \pm 0.03$ and of $E = 0.48 \pm 0.03$ (error calculation according to Eq. 1, Methods), respectively (Supplementary Fig. 2a,b and Supplementary Table 1). Apparently, under the buffer conditions used here, the nicked DNA adopts a linear conformation in solution which is stabilized by stacking interactions, similarly to that of canonical B-form DNA.

### PARP-1 binding leads to a pronounced kink in the DNA ligand containing a SSB

Binding of PARP-1 introduces a pronounced kink in the DNA and therefore leads to a shift of the observed smFRET efficiency histogram to higher values (high FRET peak at $E_{HF} = 0.87$, Fig. 1c, Supplementary Fig. 2e). Moreover, the observed histogram is asymmetric with a pronounced shoulder at lower FRET efficiencies, indicating conformational heterogeneity of the DNA-protein complex. Interestingly, when using a fragment of PARP-1 containing only its first two zinc finger domains (residues 1-214, F1F2), we observe a smFRET histogram practically indistinguishable from that in presence of the full length protein (Fig. 2, green $E_{HF} = 0.86$ see also Supplementary Fig. 2d,e and Supplementary Table 1). Thus, binding of the first two zinc fingers is sufficient for forming the fully-kinked DNA structure.

Interestingly, when using a shorter fragment of PARP-1 containing only zinc finger domain 2 (residues 103-214, F2) the recorded smFRET data still yielded a FRET efficiency histogram shifted towards higher efficiencies as compared to DNA alone, but the efficiencies are not as high as those observed in the presence of F1F2 (Fig. 2). The histogram for the F2 complex appears comparatively broad and again not entirely symmetric. This could be explained either by a remaining fraction of unbound DNA, by structural heterogeneity of the bound DNA complex, or by conformational changes on a time-scale faster than the experimental observation time (1 ms, binning time). While smFRET data provide a very sensitive tool for investigating structural changes, changes in smFRET efficiencies can also be caused by changes in the photo-physics of the dye molecules. We, therefore, performed control experiments with a slightly modified DNA construct as well as a different donor dye molecule. These experiments yielded closely comparable results (Methods, Supplementary Fig. 3), confirming our interpretation that the observed changes in smFRET efficiency histograms are caused by structural rearrangement.

Taken together, these results show that F2 alone is able to induce a kink in the DNA, but kinking is not as pronounced as that caused by binding of F1F2. In contrast, when only zinc finger domain F1 (residues 1-103, F1) is added to the DNA, the measured FRET efficiencies indicate that DNA remains in the straight conformation (Supplementary Fig. 4). Yet, comparison of the observed diffusion time to that of DNA alone indicates protein binding, although to a lesser extend than what is

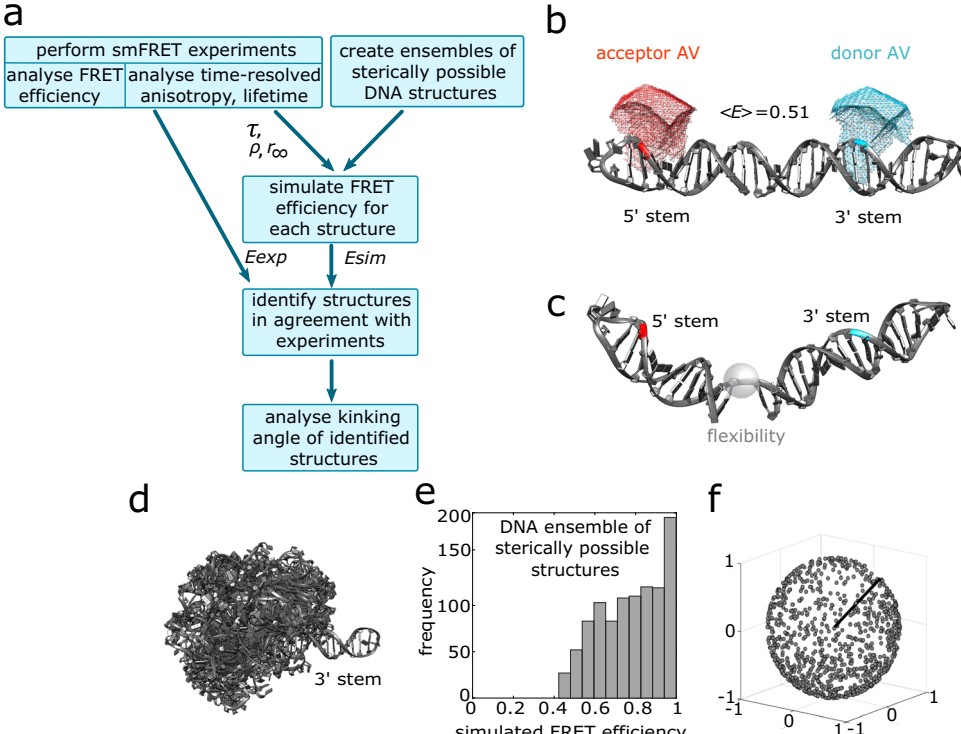

**Fig. 3 | Structural analysis of DNA conformation using a computational approach. a** Workflow for determining the kinking angles of DNA bound to different PARP-1 fragments. For an ensemble of possible DNA conformations, the expected FRET efficiencies are simulated, taking experimental data into account, i.e. Förser radius, fluorescence lifetime and time-resolved anisotropy. From the ensemble of conformations, structures are selected for which the simulated FRET efficiency is in agreement with the measured efficiency within the given uncertainties. $E_{exp}$: experimental FRET efficiency; $E_{sim}$: simulated FRET efficiency; $\tau$: lifetime; $\rho$: rotational correlation time; $r_{\infty}$: residual anisotropy. **b** Model structure for the straight DNA ligand modeled in B-form. The bases to which donor and acceptor are attached are highlighted in cyan and red, respectively. The corresponding computed accessible volumes (AV) are shown by meshes in the respective colors. **c** Example of a structure taken from the computed ensemble of sterically possible

DNA structures; the bases to which the dyes are attached are highlighted in cyan (donor) and red (acceptor), respectively. The transparent sphere represents the flexible DNA backbone in the linker region connecting the two stems. **d** Ensemble of computed model structures for the DNA ligand alone. Structures are superposed using the 3′ stem of the DNA, and the 5′ stem is allowed to adopt any sterically possible orientation relative to the 3′ stem. For visual clarity, only 100 randomly chosen structures are shown, extracted from the full set of 1000 calculated structures. **e** Histogram of simulated FRET efficiencies for all 1000 structures of the DNA ensemble represented in d. **f** Representation of all simulated DNA structures in a spherical coordinate system. All structures are aligned with respect to the 3′ stem. The axis of the 3′ stem is represented by a black line. The grey dots indicate the position of the tip of the 5′ stem for each DNA structure.

---

observed for F2 at equal concentration. This observation matches well with earlier studies showing F2 binds DNA, as well as DNA nicks with a higher affinity than F1[32,38]. Together, these data suggest that SSB recognition by PARP-1 involves binding of F2 to the DNA nick, resulting in a slightly bent structure, while additional binding of F1 resulting in a sharp DNA kink.

### Quantification of kink angles by comparing smFRET to simulated structural ensembles

Next, we wanted to quantify the degree of kinking of the DNA bound to different PARP-1 fragments, namely F2 and F1F2. While smFRET data contains distance information, such an interpretation is not straightforward, since the 1D distance information about the inter-dye distance cannot be directly converted into a 3D model structure. However, if a model structure is available for a particular conformation, computational approaches can be used to compute the expected smFRET efficiency[39–45]. This computed efficiency can then be compared to the experimentally measured FRET efficiency in order to test whether or not the structure is in agreement with the experimental data, taking into account error estimates for both the smFRET efficiency measurement and the smFRET efficiency computation. Here, we integrated such a "backwards" approach with structural ensemble calculations in order to identify, at a single-molecule level, which DNA conformations agree with our experimental smFRET data and to

analyze the distribution of kink angles in these ensembles in order to determine the extent of DNA kinking in each complex. The workflow is illustrated in Fig. 3a and explained in the following.

In order to simulate the FRET efficiency of a given structure, the first step is to calculate the accessible volume (AV) of both dyes, given the attachment points of the dyes and the geometry of the flexible linkers that attach them to the DNA (Methods)[39]. Examples of the resulting AVs are shown for the structure of straight DNA (Fig. 3b), and represent the space that the dye can access. Once the AVs are calculated, we use Markov Chain Monte Carlo simulations together with Bayesian Parameter Estimation[45] to determine the expected FRET efficiency for a given structure (Methods).

We first tested this approach using a model structure of the DNA in a linear, i.e. B-form, DNA conformation (Fig. 3b). The simulation yields a predicted FRET efficiency for this structure of $E_{sim} = 0.51 \pm 0.04$ which agrees well with the measured value of $E_{exp} = 0.50 \pm 0.03$. The given error for the simulated FRET efficiency arises predominantly from the uncertainty in the measured isotropic Förster radius of 3% ($R_{iso} = 70\,\text{Å} \pm 2\,\text{Å}$, Methods), which translates to an uncertainty of $\Delta E = 0.04$ for the simulated FRET efficiency.

The next step towards determining the kinking angle of the nicked DNA molecule was to generate a large ensemble of sterically possible DNA conformations by keeping both stems in fixed conformations while varying their relative orientation (with respect to the flexible

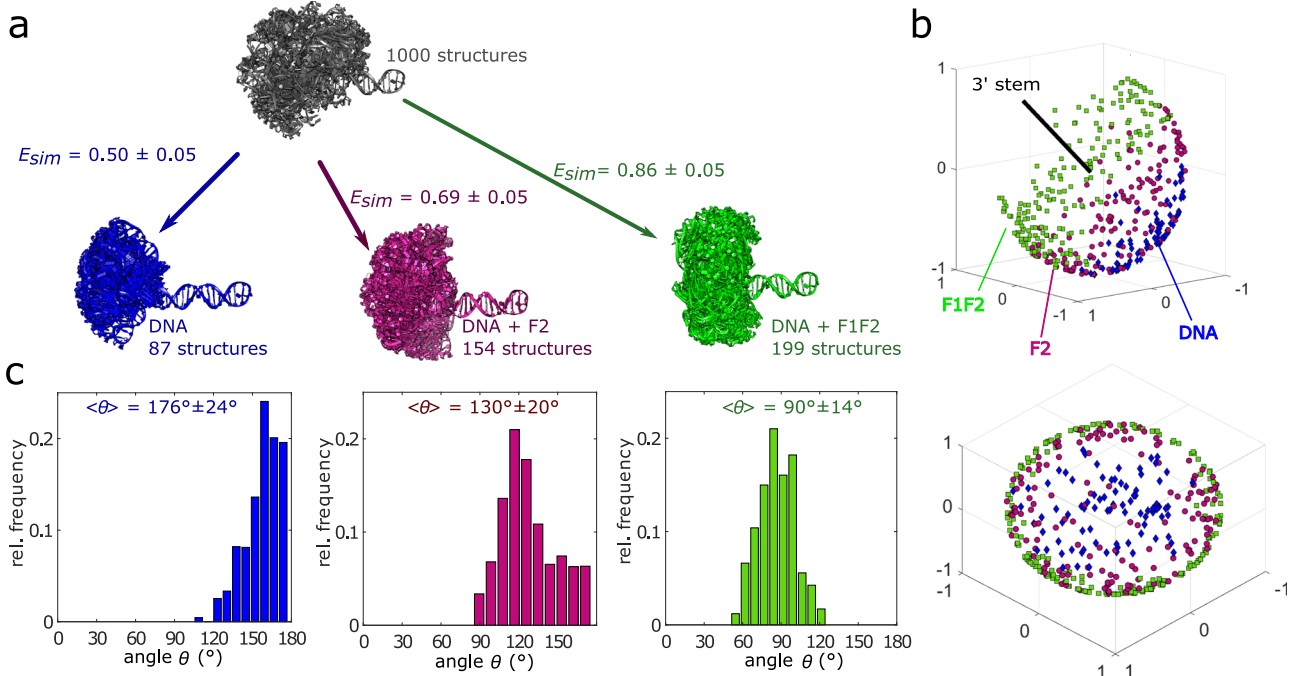

**Fig. 4 | From structural ensembles to kinking angle histograms. a** An ensemble of 1000 possible structures for the DNA was calculated (only 100 structures are shown in the figures for clarity, grey). From this ensemble, structures in agreement with the experimental FRET efficiencies were selected, i.e. 87 for free DNA (blue), 154 for DNA+F2 (magenta) and 199 for DNA+F1F2 (green). **b** Combined representation of the selected structures shown in **a**. All structures are aligned with respect to the 3′ stem. The axis of the 3′ stem is represented by a black line. The colored objects indicate the position of the tip of the 5′ stem for each selected structure (blue diamonds for free DNA; magenta circles for DNA+F2; green squares for DNA+F1F2). Two different views of the 3D plot are shown, a side view (top) and a view along the 3′ axis (bottom). **c** Kinking angle distributions for the selected structures shown in **a**. The mean and standard deviation of the angles are indicated for each histogram.

region at the nick, Fig. 3c) in any way that did not lead to steric clashes between the stems (Methods). This leads to a very heterogeneous distribution of sterically allowed DNA conformations (Fig. 3d). For each of these conformations, we then simulated the expected FRET efficiency. The resulting values show a broad distribution covering the range from $E_{sim} = 0.42$ to $E_{sim} = 1$ (Fig. 3e). Figure 3f illustrates these same DNA structures in a spherical coordinate system where the angle between the two stems (3′ stem represented by the black line and 5′ stem represented by the grey dot) was computed for every structure (Methods).

Next, in order to compare the generated structural ensembles to the recorded smFRET, we selected those structures from the ensemble that agreed with the respectively measured FRET efficiencies. To this end, for each structure we tested whether the simulated FRET efficiency was within $\Delta E = 0.05$ of the measured smFRET efficiency, where the uncertainty was obtained using error propagation, considering an error contribution of 0.03 from the experimental FRET efficiency and 0.04 from the simulated FRET efficiency (Methods). From the ensemble of 1000 structures, 87 structures were in agreement with the smFRET data for DNA alone (i.e. $E_{sim} = 0.50 \pm 0.05$, blue structures in Fig. 4a), 154 structures with the data for DNA + F2 ($E_{sim} = 0.69 \pm 0.05$, red structures in Fig. 4a) and 199 structures with the data for DNA + F1F2 ($E_{sim} = 0.86 \pm 0.05$, green structures in Fig. 4a). The respective ensembles of selected structures in Fig. 4a show that the three different FRET efficiency ranges lead to distinct distributions of kink angles θ (Fig. 4b, c). For a better overview of the selected structures, we chose a representation that displays the respective structures with reduced complexity, highlighting only the relative directions of the two DNA stems (Methods, Fig. 4b). The side view shows that the allowed kinking angle regimes are well separated for the three ensembles (upper plot in Fig. 4b). From the view along the 3′ axis (lower plot in Fig. 4b), it is apparent that the selected structures lie on

circles around the 3′ axis, indicating that our approach gives no information on the angle φ through which the 5′ stem is rotated. This is not surprising, as the structure selection is based on a single distance between the two dyes. From the histograms we determined the mean kinking angles of $\langle\theta\rangle = 176° \pm 24°$ for DNA only, $\langle\theta\rangle = 130° \pm 20°$ for DNA + F2 and $\langle\theta\rangle = 90° \pm 14°$ for DNA + F1F2 (Methods).

This kinking angle determination relies on the structural ensemble of the nicked DNA molecules. However, as a control, we also computed an alternative structural ensemble by modeling F2 bound to the 3′ stem (Methods) and again simulated the expected FRET efficiency for each structure in the ensemble (Supplementary Fig. 5). Again we compared the simulated FRET efficiencies to the experimental data given the uncertainty of $\Delta E = 0.05$ to define structural subensembles for the complex in the presence of F2 or F1F2, which are in agreement with the experimental data (Supplementary Fig. 5). From the respective structural ensembles, we again computed the mean kinking angles $\langle\theta\rangle = 127° \pm 19°$ for DNA + F2 and $\langle\theta\rangle = 87° \pm 15°$ for DNA + F1F2, which is in excellent agreement with the analysis based on the structural ensemble of the DNA only.

Moreover, in order to compare our data for the nicked DNA to the published structure for the gapped DNA, we simulated F1F2 bound to the nicked DNA, assuming the same interactions are formed. Applying the NMR constraints previously measured for the gapped DNA to the nicked DNA led to a very well-defined conformation, which is described by a structural ensemble of 40 structures (Supplementary Fig. 6a). From this ensemble, structures with predicted FRET efficiencies matching those of the experimental FRET data were selected. These selected structures have a mean angle between the two DNA stems of $\langle\theta\rangle = 102° \pm 1°$ (Supplementary Fig. 6b), in good agreement with the data for the other two computed structural ensembles. This result suggests a quite similar conformation of the complex with either a nicked or gapped DNA. One should note that NMR measurements for

F1F2 bound to a gapped DNA revealed an interaction between F1 and the unpaired base at the site of the gap itself, which is of course, not present in the nicked DNA. This can lead to different kinking angles for the two complexes. Interestingly, the simulated FRET efficiencies for the ensemble of F1F2 bound to the nick yielded a mean and standard deviation of $E_{sim} = 0.79 \pm 0.04$ (Supplementary Fig. 6c), and 18% of the 40 structures agreed with the smFRET results (i.e. $E_{sim} = 0.86 \pm 0.05$), however, these are only at the upper edge of the simulated FRET efficiency histogram. Thus our data suggests that the nicked structure is kinked to an angle slightly smaller than that observed in the NMR structure of F1F2 binding to a gapped DNA. As a control, similar angles for DNA, DNA + F2, and DNA + F1F2 were obtained for the alternative DNA construct containing different dyes (Supplementary Fig. 7), showing the robustness of the method.

## Dynamic analysis reveals fast dynamics for nick DNA in the presence of PARP-1

As described above, the observed smFRET histograms for DNA in the presence of F2, F1F2, and full-length PARP-1 showed shoulders at lower FRET efficiencies (Fig. 2). A possible reason for this could be averaging over different structural states that interconvert on a time scale on the order of 1 ms. We, therefore, used several analytical approaches to test the data for evidence of dynamics in these structures (Supplementary methods). First, we investigated the dependence of smFRET efficiency on burstwise donor lifetimes in the presence of the acceptor (Supplementary Fig. 8a). When comparing these lifetimes with the ratiometrically determined FRET efficiencies, one can analyze in two-dimensional histograms whether the observed histograms fall on the so-called static FRET line or whether deviations can be seen that are caused by dynamic interconversion of linear and kinked conformations (Methods[46]). While for the free DNA the observed 2D histogram falls on the static FRET line (Supplementary Fig. 8a), there are clear deviations observed for the DNA in the presence of F2, F1F2 or PARP-1, indicating the presence of a dynamic interconversion. Similarly, burst variance analysis (BVA[47]) also showed indications of dynamic interconversion for DNA + F2, DNA + F1F2 and DNA + PARP-1, but not for the DNA in the absence of proteins (Supplementary Fig. 8b). More direct information about the respective dynamics with time scales between 0.2 and 5 ms comes from time-window analysis, where histograms of FRET events were computed using different time-binnings (Methods, Supplementary Fig. 8c). Again, for the case of DNA alone, no dynamics were observed. Interestingly, for the DNA in the presence of protein, the resulting histograms at shorter time bins become skewed, but no distinct sub-populations become visible, indicating that dynamics occur on an even faster time-scale.

Besides actual dynamics, dye-photophysics could also influence the observed smFRET efficiency histogram. However, by using a pulsed interleaved excitation scheme[48] we were able to investigate these effects directly. To this end, we performed FRET-FCS experiments for the single molecule burst data sub-ensemble consisting of the acceptor-only species (Supplementary Fig. 9a, Supplementary methods). This analysis revealed that the acceptor, Alexa647, attached to the nick DNA in the absence or in the presence of either F2 or F1F2 has a short relaxation time of 6-7 μs (Supplementary Fig. 9b, $\tau_{phot}$). This is in good agreement with previous reports[49,50], and is presumably caused by *cis-trans* photo-isomerization, characteristic of cyanine dyes[51,52]. For the full-length PARP-1, the acceptor relaxation time is slightly longer (~10 μs) in comparison to results for the zinc finger fragments (Supplementary Fig. 9b, $\tau_{phot}$), which is most probably due to direct interactions with the additional domains in a so-called PIFE effect[53].

To provide more evidence concerning the nature of the species involved in the dynamic interconversion between the linear and kinked DNA conformations, we employed FRET-FCS experiments[54,55] (Supplementary methods). These show that the translational diffusion

times $\tau_D$ (Supplementary Fig. 10, smFRET histograms from Fig. 1c) in our confocal microscope for the low FRET population of DNA with PARP-1 (1990 ± 80 μs) are clearly distinct from that of DNA alone (1303 ± 30 μs). These diffusion times provide strong support for our interpretation that the protein remains bound to the DNA in these low-population states where the DNA is linear.

Next, we performed a quantitative analysis of the observed fluorescence dynamics caused by structural changes using filtered FCS (fFCS, Methods[56,57]). From the observed smFRET efficiency histograms of both DNA + F2 and DNA + F1F2, we selected smFRET efficiency regions towards the edge of the histogram for interconversion analysis (Fig. 5, blue and violet areas in the FRET efficiency distributions). By globally fitting the fFCS auto- and cross-correlation functions, the relaxation time for interconversion between the two respective species can be determined (Methods, Fig. 5 and Supplementary Table 2).

We find relaxation times of $\tau_R$ (DNA + F2) = 404 ± 62 μs for DNA + F2, $\tau_R$ (DNA + F1F2) = 265 ± 26 μs for DNA + F1F2, and $\tau_R$ (DNA + PARP-1) = 273 ± 19 μs for DNA + PARP-1, (Supplementary Table 2), which account for the fast dynamics in the respective systems. The relaxation time is given in each case by the inverse sum of the transition rates between the two states $\tau_R = (k_{12} + k_{21})^{-1}$, and therefore when comparing the relaxation times for DNA + F2 and DNA + F1F2 we find $(k_{12} + k_{21})_{DNA+F2} < (k_{12} + k_{21})_{DNA+F1F2}$. In summary, we observe dynamics for the DNA in the presence of F2, of F1F2, and of full-length PARP-1, and we are able to quantify the time scale of these dynamics using fFCS.

## PARP−1 inhibitors can influence the dynamics of PARP-1 at DNA damage sites

Next, we investigated whether the observed distribution of binding states can be modulated by the binding of PARP-1 inhibitors (PARPi). Some PARPi were recently shown to drive the PARP-1 allostery and activate or inhibit its function[18,58]. Here, we characterize effects caused by different PARPi of all three classes[19,18]. We observe that EB-47 traps the PARP-1-DNA complex in the kinked state, as the $E_{LF}$ (low FRET efficiency) population decreases substantially in comparison to the situation without inhibitor (Fig. 6a, pink), while olaparib shows an approximately neutral effect (Fig. 6b) in agreement with the classification. Interestingly, niraparib has a different effect on the observed smFRET histogram, since in this case the $E_{LF}$ population is increased rather than decreased (Fig. 6c). Again, FRET-FCS analysis of the low FRET peak yielded a diffusion time of 1976 ± 190 μs which is much larger than that for free DNA ($\tau_D = 1303 \pm 30$ μs, Supplementary Fig. 10) Note, a slight shift of the position of the peaks in the smFRET histograms in the presence of niraparib relative to the other observed smFRET distributions is not due to differences in the conformation, but instead is caused by the photophysics of the donor dye in the presence of niraparib (Supplementary Fig. 11).

In order to compare quantitatively the effect of the different PARP inhibitors on the dynamic distribution of PARP-1 binding states, we again turn to the quantitative analysis of dynamics using fFCS (Fig. 6, Supplementary Fig. 12 and Table 1). We observe that the "pro-retention" PARPi, EB-47, shows a decrease in the determined relaxation time, indicative of a faster rate from the linear to the kinked state. In contrast, the "pro-release" inhibitors niraparib, rucaparib and veliparib all show an increase in the relaxation time, indicative of a slower rate from the linear to the kinked state, and thus a more pronounced $E_{LF}$ peak in the FRET efficiency histogram (Fig. 6c). PARP-1 in the presence of inhibitors previously classified as class II (olaparib and talazoparib) shows an intermediate relaxation time, higher than in the presence of EB-47, slightly lower than that in the absence of inhibitors and much lower than that for class III inhibitors.

Therefore, we can summarize that different classes of inhibitors influence the kinetics between the linear and kinked DNA conformations when bound to PARP-1, and those differences can well be

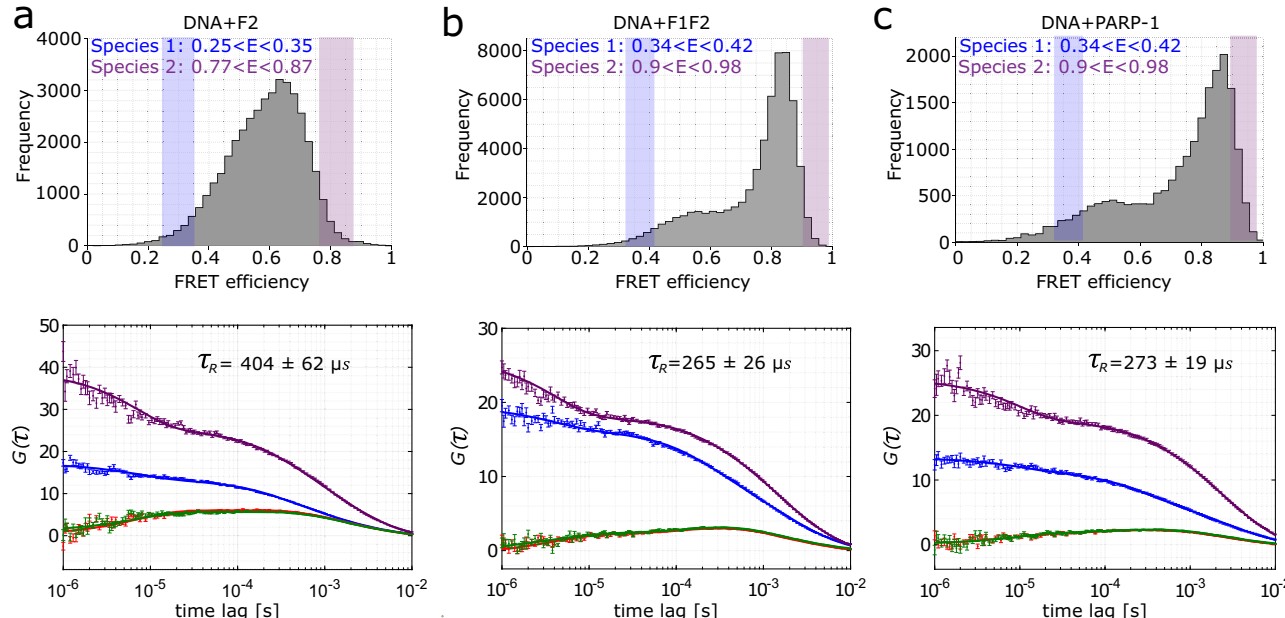

**Fig. 5 | Results of fFCS analysis for DNA in presence of F2, F1F2 and full length PARP-1.** Dynamic analysis using fFCS for DNA in presence of F2 (**a**), in presence of F1F2 (**b**) and in presence of full-length PARP-1 (**c**). FRET efficiency histograms (the same as in Fig. 2) with the thresholds that define species 1 (shaded blue region) and species 2 (shaded violet region) are shown in the upper panels and the corresponding fFCS fits for the auto- (violet, blue) and cross-correlation functions (green, red) in the lower panels. In the top right corner of the fFCS plot the resulting relaxation times are displayed. The error bars in fFCS fits show the standard error of the mean for the computed molecules contributing to the same bin.

quantified using a quantitative analysis of dynamics observed in smFRET data.

## smFRET reveals a dynamic interplay of PARP-1 and XRCC-1 at SSB sites

Next, we wanted to test whether the observed structural changes on the DNA ligand induced by PARP-1 binding have an effect on the binding of other proteins in DNA repair. Previously, the role of the binding of SSB repair factor XRCC1 in early damage recognition has been well established[11,35,59], and thus we tested whether XRCC1 binding is influenced by PARP-1-induced DNA kinking. To this end, we first performed XRCC1 DNA binding experiments in the presence and absence of F1F2 using electrophoretic mobility shift assay (EMSA, Supplementary Fig. 14). We find that F1F2-induced DNA kinking stimulates XRCC1 binding, since the unbound DNA fraction is more pronounced in the absence of F1F2 fragment, whereas in the presence of F1F2, the unbound DNA fraction is much lower and disappears (almost) completely for high XRCC1 concentrations.

Having established that PARP-1 kinking influences XRCC1 binding, we again turned to the smFRET assay to test whether additional binding of XRCC1 changes the observed kinked state of DNA when bound to PARP-1 (Supplementary Fig. 15a). However, we don't see any change in the observed smFRET distribution, and therefore no change in its conformation. Yet, we find that the diffusion time determined by FCS shows an increase in the presence of XRCC1 (Supplementary Fig. 15c), establishing the binding of XRCC1 to DNA also in the conditions of the smFRET experiments. Similarly, no effect on the FRET efficiency histogram is observed for DNA bound by F1F2 in the presence of XRCC1 (Supplementary Fig. 15b). Next, we wanted to observe whether the binding of XRCC1 changes the conformation of the DNA ligand in the absence of PARP-1 (Fig. 7a). Interestingly, again no change in the smFRET histogram is observed, while the diffusion time of the complexes increases, indicating XRCC1 binding. In contrast, in case of DNA bound by F2, where DNA is less kinked ($<\theta> = 130° \pm 20°$, Fig. 4c), binding of XRCC1 changes the observed smFRET distribution, yielding a complex multi-state profile (Fig. 7b). The data can best be described

using three Gaussians (Supplementary Fig. 16), yielding a high-FRET state with a mean FRET efficiency of $E_{HF} = 0.85$ similar to that of PARP-1 bound to the ligand ($E_{HF} = 0.86$, Supplementary Table 1). In addition also peaks at smFRET efficiencies resembling the linear and intermediate kinked DNA are observed ($E_{LF} = 0.51$, $E_{MF} = 0.7$). Interestingly, while these values are comparable to those observed for bare DNA and F2-bound DNA, single-molecule bursts selected from the respective range of the smFRET histograms are higher in the presence of XRCC1 (Supplementary Fig. 17), indicating that XRCC1 remains bound in all three DNA conformations. Thus, we find that XRCC1 binds to the prekinked DNA more efficiently than to linear DNA and binding of XRCC1 to the F2-DNA complex results also in a highly kinked state with a comparable kinking angle than what is observed for PARP-1 bound to DNA in the absence of XRCC1

## Discussion

In this study we have obtained quantitative, single-molecule insights into the mechanism of DNA single-strand breakage recognition by human PARP-1. By developing a hybrid smFRET and computational approach in conjunction with time resolved fluorescence spectroscopy analysis, we analyzed the structural heterogeneity and dynamics of nicked DNA alone, as well as when bound to full-length PARP-1 or its zinc finger domains F2 or F1F2, recapitulating the first crucial events on SBB sensing of PARP-1 activation and its interplay with XRCC-1. The free-nicked DNA adopts a conformation that resembles that of intact DNA ($E = 0.50$ for nick DNA and $E = 0.48$ for ligated DNA, Supplementary Fig. 2). It is well-known that intact DNA does not necessarily follow a strict, linear canonical B-form conformation[60,61]. Thus, the investigated DNA template may be slightly bent, however, we observe no specific effect due to the nick, and in particular, no evidence for kinking at this position. This is also consistent with the previous ensemble averaged NMR measurements of nicked DNA[62]. Using the described novel hybrid single-molecule and computational approach, we determine a mean angle of $<\theta> = 176° \pm 24°$ for the free DNA. Note, that the definition of the DNA axis, i.e. the number of base pairs considered for axis calculation (Methods), has a small impact on the

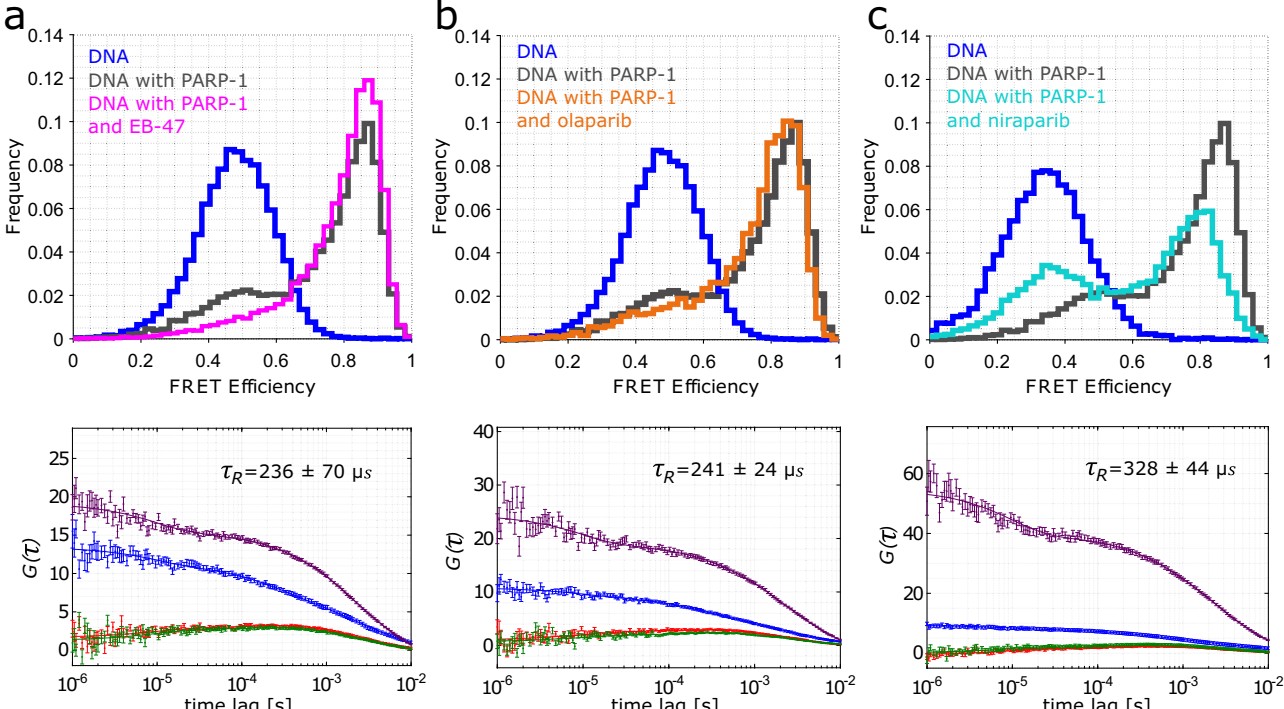

**Fig. 6 | DNA conformation probed by smFRET in presence of PARP-1 and PARP inhibitors. a** (upper panel) smFRET efficiency histogram of nicked DNA (blue) and nicked DNA in the presence of either PARP-1 (grey, same histogram as in Fig.1) or PARP-1 and EB-47 (pink), a class I PARP-1 inhibitor. **b** (upper panel) smFRET efficiency histogram of nicked DNA (blue), nicked DNA in the presence of PARP-1 (grey), as well as in the presence of PARP-1 and olaparib (orange), a class II PARP-1 inhibitor. **c** (upper panel) smFRET efficiency histogram of nicked DNA (blue), nicked DNA in presence of PARP-1 (grey), as well as nicked DNA in the presence of PARP-1 and niraparib (turquoise), a class III PARP-1 inhibitor. Note, since niraparib influences the photophysics of the donor fluorophore (Supplementary Fig. 11), unlike other PARPi (Supplementary Fig. 12), for a better comparison the DNA-only data in (**c**) is presented in the presence of niraparib. Lower panels show dynamic analysis of the respective samples using fFCS, namely DNA in the presence of PARP-1 and inhibitor: EB-47 (**a**), olaparib (**b**) and niraparib (**c**), where violet and blue are the auto-correlation and, green and red are the cross-correlation functions, respectively. In the top right corner of the fFCS plot the resulting relaxation times are displayed. The error bars in fFCS fits show the standard error of the mean for the computed molecules contributing to the same bin.

resulting angle between the axes, and the limited number of base pairs in each stem of the DNA leads to angles smaller than 180°, even for straight DNA (i.e. to θ = 176° for our model of straight DNA).

In our analysis where individual DNA structures are identified that are consistent with the smFRET data of DNA + F1F2, we found a mean angle of <θ> = 90° ± 14° (when selecting from the DNA-only ensemble, Fig. 4c) and <θ> = 87° ± 15° (when selecting from the DNA + F2 ensemble, Supplementary Fig. 5c). Taking into account the error, this result is in accord both with early electron microscopy measurements,[31] and with ensemble-averaged NMR measurements on a gapped DNA bound by F1F2[23].

In recent years, considerable efforts have been made to use smFRET measurements as quantitative tools for obtaining structural information[44,63,64]. In particular, smFRET measurements using

networks of measurements have provided structural information[40,65,66]. Here, we have shown that quantitative information from single-molecule measurements of an individual distance combined with structural modeling can be sufficient for obtaining quantitative structural models that yield direct mechanistic insight.

Our single molecule analysis of DNA structures in the presence of F2 reveals a mean angle of <θ> = 130° ± 20° (when selecting from the DNA ensemble, Fig. 4c) and <θ> = 127° ± 19° (when selecting from the DNA + F2 ensemble, Supplementary Fig. 5c), which is a clearly distinct result from those for either free DNA (176° ± 24°, Fig. 4c) or DNA + F1F2 (90° ± 14°, Fig. 4c). Binding of F2 alone leads to a kink in the DNA that becomes significantly sharper in the context of binding by F1F2. Importantly, the sharp kinking angles of the nicked DNA template seen in the F1F2 complex are neither observed for the DNA alone, nor in the presence of just F2, in agreement with an induced fit mechanism for DNA damage recognition by PARP-1. Moreover, even though we do observe dynamics, results from the fFCS analysis rule out a conformational selection mechanism. The conversion time for DNA + F2 determined from the fFCS analysis ($\tau_R$ (DNA + F2) = 404 ± 62 µs) is longer than that for DNA + F1F2, ($\tau_R$ (DNA + F1F2) = 265 ± 26 µs), and therefore the sum of the interconversion rates is smaller. In a conformational selection model, the forward rate for formation of the partial kink by binding of F2 ($k_{12}^{F2}$) would have to be higher than or equal to that for the formation of the full kink by binding of F1F2 ($k_{12}^{F1F2}$), since thermal fluctuations would cause bending to an angle required for F2 binding more often than to the sharper angle required for F1F2 binding. Moreover, when comparing the smFRET histograms of DNA + F2 and DNA + F1F2 (Fig. 2, Supplementary Fig. 2), the DNA + F1F2 histogram has two local maxima at about 0.6 ($E_{LF}$) and 0.86 ($E_{HF}$)

**Table 1 | Interconversion times obtained from fFCS analysis for DNA in presence of PARP-1 and PARPi**

|           | Dataset                              | $\tau_R$, µs |
|-----------|--------------------------------------|--------------|
|           | DNA with F2                          | 404 ± 63     |
|           | DNA with F1F2                        | 265 ± 26     |
|           | DNA with PARP-1                      | 273 ± 19     |
| Class I   | DNA with PARP-1 and EB-47            | 236 ± 70     |
| Class II  | DNA with PARP-1 and olaparib         | 241 ± 24     |
|           | DNA with PARP-1 and talazoparib      | 245 ± 88     |
| Class III | DNA with PARP-1 and niraparib        | 328 ± 44     |
|           | DNA with PARP-1 and veliparib        | 338 ± 60     |
|           | DNA with PARP-1 and rucaparib        | 365 ± 36     |

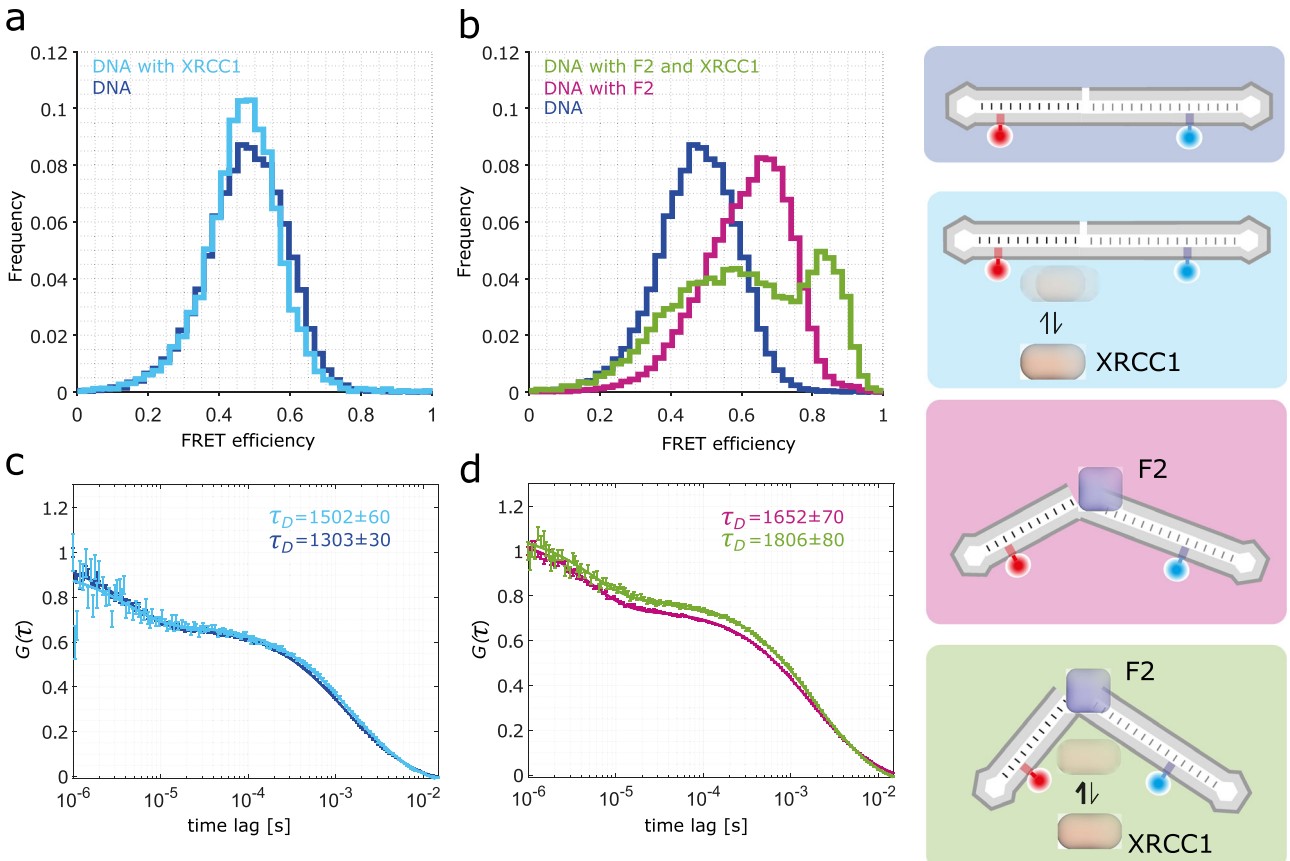

**Fig. 7 | smFRET measurements probing the DNA conformation in presence of XRCC1. a** smFRET efficiency histogram of nicked DNA (blue) and nicked DNA in presence of 1 µM XRCC1 (light blue); **b** smFRET efficiency histogram of nicked DNA (blue), nicked DNA in presence of 10 µM F2 (magenta) and nicked DNA in presence of 10 µM F2 and 1 µM XRCC1 (dark green). **c, d** FRET-FCS results for data in a and b, respectively, using the same color coding; the extracted diffusion times are displayed in the upper right corner of the FRET-FCS plot and complete fit results are shown in Supplementary Table 6. The error bars show the standard error of the mean the computed molecules contributing to the same bin.

FRET efficiency, with the higher efficiency state being the preferred state, while the F2 histogram shows only one, broader peak. Given the significant difference between the two states in the DNA + F1F2 histogram, the backward rate to the linear conformation, $k_{21}^{F1F2}$, of DNA + F1F2 would have to be smaller than $k_{21}^{F2}$ if the peak observed for F2 were to be due to an averaging of different states within the conformational search, in complete contrast to the fFCS data actually measured. Therefore, we can conclude that the observed state in the presence of F2 is a structurally independent state, thereby ruling out the conformational selection model. Instead, protein binding leads to progressive kink formation at each step, as expected for the induced fit mechanism.

It has been suggested that cooperative binding of both N-terminal PARP-1 zinc fingers constitutes the crucial event not only for SSB recognition, but also for triggering the domain assembly cascade leading to PARP-1 activation[23,67]. Our single molecule data now provides direct mechanistic insights into SSB recognition by showing that F2 induces DNA kinking and makes the 3′ stem binding site accessible for interaction with F1. It is important to point out, that F2 by itself is not sufficient for DNA damage recognition. It needs the additional binding of F1 in order to form the completely kinked structure. This is in accordance with the observation that in live-cell experiments, recruitment of F1F2 to the sites of DNA damage is comparable to full-length PARP-1, while recruitment of F2 alone is much weaker[23]. Interestingly, however, the two zinc fingers have different functionality, as shown by the fact that F1 has a lower affinity for nicked DNA structures[32,38], F1 recruitment to DNA damage in living cells was not detectable[23], and that F1 binding to DNA does not introduce DNA

bending (Supplementary Fig. 3). The data is best explained by an F2-induced opening of the DNA that twists the two DNA stems away from each other followed by binding of F1 to the exposed 5′ stem as well as formation of the interdomain interactions between F1 and F2 which drives the DNA into its highly kinked conformation. Hence, F2 and F1 work together in a sequential manner using an induced fit mechanism for allowing stable F1 binding. Similarly, our data of XRCC1 binding to DNA does not lead to an observable change of DNA conformation, however, in combination with F2 a distribution of different DNA conformations are observed, also including the fully kinked state.

This induced fit mechanism of DNA damage recognition by PARP-1, involving a multi-domain assembly and concerted DNA perturbations, differs from those of other damage sensors, where DNA perturbations are coupled to an active process using energy derived from ATP-hydrolysis (e.g. as in the case of MutS,[1]) or to conformational selection by a relatively rigid recognition surface (e.g. as in the case for the UV-damaged DNA-binding protein DDB2,[9]). It partially resembles the induced fit described for Fen1, in which loops and helical elements of the cap become ordered upon DNA kinking by the otherwise relatively rigid recognition module[8]. The cooperative behavior of the two N-terminal zinc fingers of PARP-1 helps to explain the specifity of SSB sensing, while it is also consistent with their mutual requirement for efficient recruitment to laser induced DNA lesions in cells[23,68]. Interestingly, the revealed pathway for an induced fit DNA damage recognition mechanism also helps to understand PARP-1 activation. The recognition of the 3′ stem followed by F1 binding to the 5′ stem of an SSB has been shown to trigger the cascade of domain-domain interactions that leads to activation of PARP-1's C-terminal catalytic

domain[23], explaining previous data that F2 is required for PARP-1 activation by an SSB, whereas it is dispensable for recognition of a DSB, where the F1 binding site is not obscured by a second stem[69].

Moreover, even though the amplitude of the observed high FRET peak is quite high, analysis of the observed dynamics in the smFRET data shows that even the highly kinked DNA state is not completely stable, and transient excursions to an (almost) linear DNA conformation are revealed by the fFCS analysis. In this context, we note that a linear DNA conformation of DSBs has also been observed for the related PARP-2 protein in the presence of nucleosomes[70,71]. PARP-2 contacts DSBs via its WGR domain. Given that the WGR domain of PARP-1 has been implicated in DNA intersegment transfer during damage search[13,72] it is conceivable that a similar configuration can be also adopted by PARP-1 in context of SSBs or intact DNA. In addition, the BRCT domain of PARP-1 has recently been shown to make contact with linear intact DNA[73], possibly also contributing to the stabilization of the linear state shown here.

PARP-1 is an important clinical target in the treatment of cancer. Here, PARP inhibitors are used to trap PARP-1 at DNA breaks. Recently, using hydrogen exchange, three classes of inhibitors have been defined based on their allosteric action; while class I inhibitors, such as EB-47 stabilize the bound conformation, class III inhibitors weaken the interaction with the DNA damage and class II inhibitors such as Olaparib show intermediate behavior[18,58]. Here, our single molecule study provides novel mechanistic insight by directly monitoring structural changes of DNA SSBs in response to the allosteric effect of PARP-1 inhibitors. For the case of EB-47, the observed effect matches our expectations. The kinked DNA conformation is stabilized, and the low FRET state for the linear conformation virtually disappears (Fig. 6a). Interestingly, class III PARP-1 inhibitors such as niraparib have the opposite effect, the kinked state is de-stabilized as predicted by the hydrogen exchange data[18] and the linear conformation becomes more prominent (Fig. 6c). While the observed smFRET values could indicate that niraparib causes the dissociation of PARP-1, this is not consistent with other evidence. In the highly dilute situation of the single molecule experiment, re-binding of PARP-1 would be fairly slow and one would expect the equilibrium to shift even more to the side of the low FRET peak in a PARP-1 concentration-dependent manner. However, direct evidence, that the observed low FRET population in the presence of niraparib is due to a linear conformation of PARP-1 bound to the nicked DNA comes from a FRET-FCS analysis of the low FRET population (Supplementary Fig. S8). Here, the observed diffusion time of $1980 \pm 190\,\mu s$ matches that of the DNA + PARP-1 in absence of inhibitor ($1990 \pm 80\,\mu s$) and is clearly distinct from that of free DNA ($1300 \pm 30\,\mu s$), showing that the inhibitor does not lead to an increased unbinding but rather to an increase in the population of a linear DNA conformation bound to PARP-1. This also is in agreement with the observation that niraparib does not change the unbinding rate of PARP-1 when bound in the context of nucelosomal substrates[13]. Having established that the observed binding state is a actually described by a dynamic interchange between linear and kinked conformation we compared the effect of different inhibitors by their influence in the dynamic interconversion between these two states. Dynamic analysis of several class I, II, and class III PARP inhibitors show distinct relaxations times, indicating that the observed dynamic interconversion between kinked and linear states is critically important for the function of PARP-1 and its inhibition by chemical ligands. These reported differences together with known structures of inhibitors bound to the active site, are likely to become important for understanding the mechanistic differences of different PARP inhibitors underlying different reported potencies of these inhibitors[16,28,74–76].

Increasing evidence indicates that the role of PARP-1 at DNA damage sites is not limited to signaling via the formation of PAR chains upon activation, but may also involve a direct function of its dynamic multi-domain structure in concert with other DNA damage response and chromatin factors. Our data reveal that recognition of the kinked DNA damage site by XRCC1 is stimulated by the presence of PARP-1. Recruitment of XRCC1 to DNA damage sites was previously shown to depend on the activation of PARP-1[68]. However, our data reveal that DNA binding of XRCC1 on its own does not alter the conformation of the DNA ligand, whereas XRCC1 binding to DNA in the presence of F2 changes the equilibrium of states resulting, at least partially, in a population of a fully kinked DNA conformation, similar to that observed for full-length PARP-1. Consequently, we conclude that DNA kinking during SSB sensing by PARP-1 may play a multi-faceted role: it activates PARP-1 by triggering its domain assembly cascade, while it can also stimulate an interplay with other DNA repair factors downstream of PAR signaling, such as XRCC1.

Interestingly, XRCC1 has been recently shown to counteract PARP-1 trapping at DNA damage sites in cells[37]. Moreover, a comparison against structures of other complexes suggests that repair enzymes such as DNA polymerase β or Fen1 recognize also a highly kinked DNA conformation, but that their binding mode may be, in contrast to XRCC1, mutually exclusive with that of PARP-1 (Supplementary Fig. 18). Taken all of this together, it is tempting to speculate that DNA kinking induced by PARP-1 could kinetically "hand-off" the recognized lesion to XRCC1 in order to enable its orchestration of subsequent repair of the lesion by the DNA repair core machinery, a mechanism that could be tested by future experiments.

In summary, we used time-resolved smFRET measurements and a computational approach to investigate the role of the zinc fingers F1 and especially F2 in recognition of DNA nicks by PARP-1. Our results show that F2 alone is able to introduce a kink at the nick, which is not present in the isolated DNA. We also show that further kinking at the nick occurs on binding to F1F2. These findings support a model in which F2 opens the DNA at an SSB and makes the binding site for F1 accessible, but only the binding of both zinc fingers together leads to the highly kinked conformation, providing a possible pathway for DNA damage recognition as well as the first crucial steps for catalytic activation of PARP-1 via successive domain assembly. The quantitative analysis of the observed smFRET dynamics using fFCS allows us to rule out a conformational selection model and provides strong evidence for an induced fit mechanism. Interestingly, smFRET analysis shows the presence of an additional, minor linear DNA conformation when bound to PARP-1 and a dynamic interconversion between major (kinked) and minor (linear) states. The equilibrium between these two structural states is altered by PARP-1 inhibitors. Future studies could address whether the linear state represents an intermediate in the unbinding pathway of PARP-1 and how other proteins of the repair machinery alter the observed dynamic equilibrium e.g. during the hypothesized hand-off to XRCC1.

## Methods
### Preparation of DNA ligands
The double labeled single-stranded DNA (sequence 5′-GCTGGCTG ATCGTAAGATCAGCCAGCCGCGGTCGGCTGTCAGCTTGCTGACAGCC GACCGCG-3′, with Alexa647 attached to T18 via a C2 linker and ATTO550 attached to T37 via a C2 linker, referred as DNA$_{ATTO}$, Fig. 1b) was received in two parts which were ligated to produce the final dumbbell like structure. The acceptor strand (sequence 5′-GCTGGCTGATCGTAAGATCAGCCAGCCGCGGTCG-3′ with Alexa647 attached to T18 via a C2 linker, was synthesized by IBA GmbH (Göttingen). The donor strand with sequence 5′-GCTGTCAGCTTGCTGA-CAGCCGACCGCG-3′ with an amino C2 modification at T26 was synthesized by Biomers (Ulm). For labeling of the donor strand, the lyophilized oligos were dissolved in TE buffer (10 mM Tris pH8.0, 1 mM EDTA) and then mixed 1:1 with a 100 mM sodium tetraborate buffer pH 8.5 to a final DNA concentration of $100\,\mu M$ in $50\,\mu l$. The labeling reaction was started by adding 50 nmol ATTO 550-NHS-Ester (Sigma-Aldrich, 92835, dissolved in DMSO). After incubation for 1 h at 37 °C,

another 50 nmol of dye were added and incubated overnight at 37 °C. The labeled DNA was then separated from free dye and unlabeled DNA by denaturing PAGE and gel extraction. The donor-labeled DNA was phosphorylated at the 5′ end using T4 Polynucleotide Kinase (New England BioLabs, M0201S) for 1 h at 37 °C, followed by heat inactivation. To ensure the nick formation in the final construct, the acceptor strand DNA was dephosphorylated at the 5′ end using rSAP (New England BioLabs, M0371) for 1 h at 37 °C, followed by heat inactivation.

For ligating the donor and acceptor strands, both were mixed (1 μM acceptor strand and 1 μM phosphorylated donor strand in a final volume of 19 μl) in T4 DNA ligation buffer (New England BioLabs, B0202S) supplemented with 1 mM ATP (New England BioLabs, P0756S). The strands were annealed by heating to 90 °C for 30 seconds followed by an incubation for 4 minutes at 42 °C, decreasing the temperature by 1 °C per minute down to 21 °C and incubating again for 4 minutes. After annealing, the sample was ligated overnight at 16 °C using T4 DNA ligase (New England BioLabs, M0202S). The final construct has two tetraloops on either sides of the nick to minimize unspecific PARP-1 binding[32].

Another DNA construct with identical sequence but with Alexa647 attached to T18 via a C6 linker and 6-Tamra attached to T37 via a C6 linker (referred as DNA$_{Tamra}$) was synthesized by IBA GmbH (Göttingen). The desalted DNA was purified and folded into the dumbbell structure as previously described[32].

## Phosphorylation and ligation of nicked DNA ligand
The double-labeled DNA ligand was phosphorylated using the T4 Polynucleotide Kinase for 1 hour at 37 °C, followed by heat inactivation. This 5′-phosphorylated version of the DNA ligand was used for all smFRET experiments.

For ligation of the nick, the phosphorylated DNA was diluted to 250 nM in T4 DNA Ligase Buffer and 2000 units of T4 DNA ligase were added in the reaction volume of 20 μl. The reaction was incubated overnight at 16 °C and then the ligase was heat inactivated.

## Protein expression and purification (PARP-1, F1F2, F2, XRCC-1)
The codon optimized DNA plasmid for PARP-1 expression in *E. coli* was kindly provided by David Neuhaus. Expression and purification procedures were adopted with minor changes from protocols also provided by Sebastian Eustermann and David Neuhaus and protocols from[77]. Expression of PARP-1 was performed using *E. coli* BL21 (DE3) cells (Novagen) in Lysogeny Broth (LB) media supplemented with 50 μg/ml kanamycin and 10 mM benzamide. All the colonies from a transformation plate were transferred to 60 ml LB$^{hiBenz}$ media supplemented with 0.5% Glucose, 2 mM MgSO4, 50 μg/ml Kanamycin to grow the starter culture to ≥2.0 OD$_{600}$. A total of $3 \times 1$ l of LB$^{hiBenz}$ were inoculated with 1:50 starter and allowed to grow to 0.8–1.2 OD$_{600}$, 37 °C, 200 rpm. Cell growth was arrested at 0.7 OD$_{600}$ and incubated for 1 h at 4 °C. Afterwards, expression was induced with 0.5 mM IPTG and 0.15 mM ZnSO4 and allowed to proceed overnight at 25 °C, shaking at 200 rpm. After centrifugation, the cell pellet was resuspended in 80 ml buffer containing 25 mM HEPES-Na, pH 8, 0.5 M NaCl, 0.5 mM Tris(2-carboxyethyl)phosphine hydrochloride (TCEP), 1 mM PMSF and 1 protease inhibitor tablet (EDTA free, Sigma) and sonicated for 3 minutes at 75% amplitude, 3/8 seconds on/off. After clearing the lysate, it was mixed with 8 ml Ni-NTA beads (50% slurry, Qiagen) and incubated for 90 min on rollers at 4 °C. The lysate mixture with Ni-NTA agarose was applied to a Bio-rad gravity column (2.5 x 10 cm). After the flow through was collected, the Ni-NTA agarose with bound PARP-1 was washed with 50 ml LSW (low salt wash buffer containing 25 mM HEPES-Na, pH 8, 0.5 M NaCl, 20 mM Imidazole, 1 mM PMSF, 0.5 mM TCEP) followed by 50 ml HSW (high salt wash buffer as LSW but with 1 M NaCl) and again with 50 ml LSW. The elution was performed using 10 ml, 8 ml, and 8 ml with 5 minutes incubation of elution buffer containing 25 mM HEPES-Na, pH 8, 0.5 M NaCl, 400 mM Imidazole, 1 mM PMSF, 0.5 mM TCEP. After purification with Ni-NTA agarose the protein was diluted to 375 mM NaCl and further purified on an ÄKTA purifier system (GE Healthcare Life Sciences) at 4 °C using 1 ml HiTrap™ Heparin HP column (Cytiva). The protein was eluted in a buffer containing 50 mM Tris, pH 7, 0.5 mM TCEP, and a salt gradient from 375 mM to 1 M NaCl. After this purification steps the PARP-1 was subjected to buffer exchange (20 mM Hepes pH 8.0, 200 mM NaCl, and 0.1 mM TCEP) with Zeba™ Spin Desalting Columns, 7 K MWCO, 0.5 ml (Thermo Scientific, 89882).

Expression and purification of F1 and F2 was performed as described[32], with the following minor modifications: Expression was carried out in Rosetta2 (DE3) cells (Novagen) and at 16 °C; one tablet of SigmaFast-EDTA free protease inhibitors was used during expression, but no protease inhibitor was added during purification; ion exchange chromatography for initial protein purification was performed on a 1 ml HiTrap SP FF (GE healthcare) and a 24 ml Superdex 75 10 300 GL (GE healthcare) was used for size-exclusion chromatography. The respective protein sequences are given in Supplementary Table 3.

hXRCC1 protein was expressed and purified according to a published protocol[78] with some modifications. hXRCC1 protein with 6-HIS tag at the C-terminus was expressed from the plasmid pET28-hXRCC1 (Addgene #70759) in BL21 (DE3) cells (Novagen). Expression was induced at OD$_{600}$ 0.6 by the addition of IPTG to 1 mM. After 90 minutes at 37 °C cells were harvested, cell pellets flash frozen in liquid nitrogen and kept at -80 °C till further use. Cells were resuspended in sonication buffer (50 mMHepes, pH8, 500 mM NaCl, 0.1 mM EDTA, 10 % glycerol), supplemented with 1 mM imidazole, 1 mM DTT and 1 mM PMSF. Cells were lysed by sonication and cellular debris removed by centrifugation (14 000 g, 20 min, 4 °C). The cleared lysate was loaded on the 5 ml HisTrap column (GE Healthcare) at the flow rate 1 ml/min. The column was washed with 6 column volumes (CV) of sonication buffer supplemented with 1 mM imidazole, 6CV of wash buffer (50 mM Hepes-NaOH, pH 7.0, 0.1 M NaCl, 0.1 mM EDTA, 1 mM DTT, 10% glycerol) supplemented with 11 mM imidazole and 5CV of wash buffer supplemented with 41 mM imidazole at a flow rate of 1 ml/min. Proteins bound to the column were eluted with the increasing concentration of imidazole (41–250 mM gradient). Collected fractions were analysed by SDS-PAGE, fractions containing hXRCC1 protein pooled and loaded on the SourceS column (GE Healthcare). Protein fractions eluted from the column with the salt gradient (100–2000 mM NaCl) showed the presence of nucleic acids, so peak fractions containing hXRCC1 proteins were reloaded on the 5 ml SP HiTrap column (GE Healthcare) and eluted with the salt gradient (100–2000 mM NaCl). The purity of the collected protein was analysed by the SDS-PAGE and its identity confirmed by mass spectrometry. Freshly purified protein was used for further experiments.

## Single-molecule FRET measurements
SmFRET measurements were performed on a custom-built confocal fluorescence setup using time-correlated single photon counting (TCSPC) and pulsed interleaved excitation combined with multi-parameter fluorescence detection (PIE-MFD)[79]. The setup has been described in detail[80].

In brief, excitation lasers at 531 nm and 640 nm were pulsed with a repetition rate of 20 MHz and with a shift of about 20 ns between each other. The excitation light is focused into a sample droplet by a 1.2NA water immersion objective. Excitation powers were 40 μW for the red laser and 95 μW for the green laser for measurements of the DNA labeled with ATTO550 and Alexa647. For measurements of the alternative DNA labeled with 6-Tamra and Alexa647, intensities were 25 μW for the red and 95 μW for the green laser. Light emitted from single molecules diffusing through the confocal volume was collected by the same objective, focused onto a 75 μm pinhole, and split into four detection channels according to polarization and spectral range. The light in each channel was focused onto a single-photon-counting

avalanche photodiode, and photon arrival times were recorded by TCSPC electronics. Single measurements were recorded for 30 minutes with a maximum of 13 molecules per second diffusing through the confocal volume.

Prior to smFRET measurements with DNA + F1F2, DNA + F2, DNA + F1 or DNA alone, the DNA was diluted in measurement buffer (12 mM HEPES pH 8.0, 60 mM KCl, 3 mM MgCl$_2$, 50 μM ZnSO$_4$, 4% Glycerol, 0.5 mM DTT) to a concentration of 5-30 pM. In order to obtain the optimal smFRET histograms, for F2 and F1F2 constructs we performed titration experiments monitoring the resulting smFRET data (Supplementary Fig. 19). Then, the respective PARP-1 fragment was added to a final concentration of 1 μM in case of the DNA + F1F2 or 10 μM in case of the DNA + F2 or DNA + F1 and then used for measurements directly. For smFRET measurements with DNA + PARP-1, 20 pM of DNA in presence of 1 μM protein in an optimized buffer (20 mM HEPES pH 8.0, 200 mM NaCl, 0.1 mM TCEP, see Supplementary Fig. 20 for details about buffer optimization) was used. The measurements with PARPi were performed in the same conditions as with PARP-1 but with addition of 200 μM of either EB-47, olaparib, talazoparib, niraparib, rucaparib or veliparib (MedChemExpress LLC, USA). PARPi stock solutions were prepared in DMSO such that the stock concentration allowed the DMSO percentage in smFRET measurements to be kept below 2%. All samples were kept on ice until start of the measurements, which took place at room temperature. A droplet of the sample was placed on a PEGylated coverslip (Marienfeld No. 1) and restrained using Roti Liquid Barrier Marker, colorless (Roth).

For PEGylation of coverslips, they were first cleaned by cooking once in a 2% solution of Hellmanex (Hellma Analytics) and twice in water. Afterwards, the coverslips were rinsed with water and dried under a nitrogen stream. The dry coverslips were treated in a PlasmaCleaner (Zepto, Diener electronic) for 10 minutes with oxygen plasma at 80% power. The clean coverslips were silanized by incubating in a solution of 2% (v/v) 3-aminopropyl-triethoxysilane (Sigma-Aldrich A3648) in acetone for 30 minutes. After washing and drying, the coverslips were PEGylated by incubation with a solution of mPEG-SVA (methoxy-poly(ethylene glycol)-succinimidyl valerate (Laysan Bio Inc. #MPEG-SVA-5000, 400 mg/ml in a 3:7 carbonate:bicarbonate buffer pH 9.4) for 45 minutes. PEGylated coverslips were rinsed with water, dried under nitrogen stream and kept in a dry container until use[81].

## Data analysis

Data analysis was performed using a MATLAB based software package called PAM (PIE analysis with MATLAB)[82]. The newest version is available via a repository (https://gitlab.com/PAM-PIE/PAM).

Single-molecule events (bursts) were identified in the photon time traces by applying an all-photon burst search[83], requiring at least 10 photons in 500 μs and more than 50 photons in total per burst. Files with more than 13 bursts per second were not considered for analysis to avoid frequent occurrence of multi-molecule events.

All data sets were corrected for background, direct excitation ($\delta$), crosstalk ($\alpha$), $\gamma$ (accounting for differences in quantum yield and detection efficiencies for the two dyes) and $\beta$ (accounting for different absorption cross sections and laser intensities), following standard procedures[63]; the values determined are summarized in Supplementary Table 7. Background count rates were determined from a measurement of pure buffer and ranged from 0.1 to 0.9 kHz, depending on polarization and color of the detection channel. Crosstalk was determined from the donor-only labeled species in the measurements. Direct excitation was quantified from an acceptor-only reference, comprising only the 5′ stem of the DNA$_{ATTO550}$. $\gamma$ and $\beta$ were obtained from a fit of the stoichiometry vs. FRET histograms of high and low FRET species (DNA with and without protein, respectively), as described in[79,84]. For calculation of burst-wise anisotropies of the acceptor, further correction factors were taken into account, namely

$l_1 = 0.03$ and $l_2 = 0.09$, accounting for depolarization introduced by the high-NA objective[85] and $G = 0.95$, accounting for different detection efficiencies of the parallel and perpendicular polarization detection channels[86]. Burst-wise lifetimes were calculated based on a maximum likelihood estimator[87].

FRET events were isolated from the data using an upper threshold of 10 for the ALEX-2CDE filter described in[88], which removes multi-molecule and blinking events. The remaining donor- or acceptor-only events were removed with a stoichiometry threshold ($0.3 < S \text{ to } < 0.75$). The presented FRET histograms contain only events with at least 100 photons with a duration cut at 20 ms and a corrected FRET efficiency between 0 and 1. The mean FRET efficiencies for each DNA conformation were quantified by fitting Gaussian functions to the peaks. The amount of events for each data set, before and after the application of above-mentioned filters, are presented in Supplementary Table 8.

The uncertainty for the experimental FRET efficiency was determined by error propagation of the uncertainties in the correction factors and background counts according to Eq. 1.

$$\Delta E = \sqrt{(\Delta E(\gamma))^2 + (\Delta E(\alpha))^2 + (\Delta E(\delta))^2 + (\Delta E(B_{DD}))^2 + (\Delta E(\delta B_{DA}))^2 + (\Delta E(\delta B_{AA}))^2} \quad (1)$$

In analogy to[63], the contributions can be calculated by Eqs. 2–7.

$$\Delta E(\gamma) = (1 - E) \cdot E \cdot \frac{\Delta \gamma}{\gamma} \quad (2)$$

$$\Delta E(\alpha) = (1 - E)^2 \cdot \frac{\Delta \alpha}{\gamma} \quad (3)$$

$$\Delta E(\delta) = (1 - E) \cdot \beta \cdot \Delta \delta \quad (4)$$

$$\Delta E(B_{DD}) = [\gamma \cdot E + (1 - E) \cdot \alpha] \cdot \frac{\Delta B_{DD}}{F} \quad (5)$$

$$\Delta E(B_{DA}) = (1 - E) \cdot \frac{\Delta B_{DA}}{F} \quad (6)$$

$$\Delta E(B_{AA}) = (1 - E) \cdot \delta \cdot \frac{\Delta B_{AA}}{F} \quad (7)$$

The relevant correction factors are summarized in Supplementary Table 7. Following[63], we used relative errors of 10% for $\alpha$, $\beta$, and $\gamma$. For the mean photon number per burst $F$, we used 75 and an error in the background counts of 1 photon ($\Delta B_{DD}$, $\Delta B_{DA}$, and $\Delta B_{AA}$, for donor, FRET, and acceptor channel, respectively). The resulting error $\Delta E$ depends on the measured FRET efficiency and ranges from 0.01 to 0.03 for the relevant range of FRET efficiencies, so we used $\Delta E = 0.03$ as the upper estimate.

## Generation of model ensembles

Models were calculated with the program XPLOR-NIH[89] using a slightly adapted version of the procedure previously described in detail in[23] Supplemental Experimental Procedures. First, template structures were calculated for the DNA and protein components. The protein template structures were calculated starting from the deposited co-ordinates of PARP-1 F1 and F2 domains in complex with DNA blunt ends; chains B, I, and J from pdb 3ODA[38] were used for the complex of F1, and chains B, E, and F from pdb 3ODC[38] were used for the complex of F2. These co-ordinates were adapted to the XPLOR-NIH force field by the addition of hydrogen atoms followed by energy minimization while holding all the backbone peptide groups fixed. A starting DNA dumbbell template structure was generated by taking the lowest energy structure from an ensemble of 50 simulated annealing

structures calculated using a combination of backbone intra-strand and inter-strand distance restraints, base pairing hydrogen-bond distance restraints, and dihedral angle restraints for ideal B-form geometry in the stems as well as weak base-pair planarity restraints, and using distance and dihedral angle restraints corresponding to measurements from RNA tetraloop structures 1MSY and 1RNG for the first and second tetraloops respectively (as in the previous work, the ring of the second T in the second tetraloop was modeled in the *anti* conformation). The resulting DNA dumbbell structure served as the template for ensembles containing DNA only. For ensembles that included, in addition to the DNA, either F2 alone (bound to the DNA 3′ stem) or both F1 and F2 (bound to the 5′ and 3′ DNA stems, respectively), those DNA stems that were to be bound to protein were adapted to mimic very closely the conformations found in the corresponding DNA stems in 3ODA (for the 5′ stem) and 3ODC (for the 3′ stem). This was achieved by simulated annealing calculations during which the ideal B-form backbone-backbone distance restraints and dihedral angle restraints in the 7-basepair regions bound by protein were replaced by corresponding restraints using values measured in 3ODA or 3ODC; during this process, the remaining portions of each dumbbell were restrained using the same ideal B-form restraints within the stems and tetraloop restraints as described above. The only adaptations required for the present calculations relative to those described in[23] comprised the use of these B-form restraints in stem regions between the protein binding site(s) and the tetraloops, required here because the stems being modeled were longer, as well as the absence of the single nucleotide linker that had been present between the 3′ and 5′ stems in the earlier calculations.

Ensembles were generated using these template structures by first randomizing the five rotatable dihedral angles of the 3′ stem to 5′ stem linker in the DNA dumbbell, then adding the template protein structures for those stems that were to be protein-bound, using for fitting the DNA backbone atoms in the 7-basepair region of each protein domain's DNA-binding footprint that were in common (these atoms had also been taken from 3ODC and 3ODA during generation of the protein templates); in this way, each protein domain was carried accurately into the same spatial relationship to the appropriate stem of the dumbbell as it had with blunt-ended DNA in complex 3ODA (for F1) or 3ODC (for F2). Structures were then subjected to long simulated annealing calculations with small step sizes, during which the conformation of the protein domains, the DNA stems, and tetraloops, but crucially not the DNA linker, were restrained using non-crystallographic symmetry (NCS) restraints relative to rigidly constrained copies of the starting template structures; this procedure is described in detail in[23] Supplementary methods. In the case of the F1F2-DNA ensemble, the exact same NMR-derived restraints were applied during this annealing process as had been used in[23], that is a combination of NOE-derived interdomain and intermolecular distance restraints and residual dipolar coupling-derived orientational restraints as measured for the experimentally determined complex of F1F2 with a 45-nucleotide dumbbell. In the cases of the F2-DNA ensemble and the DNA-only ensembles, no NMR-derived restraints were applied during the annealing calculations. For the F1F2-DNA ensemble, 500 structures were calculated, from which the best 49 were selected according to three simultaneously applied selection criteria based on XPLOR-NIH energy terms (these criteria were E(total) <6000 kcal.mol⁻¹, E(NOE) < 30 kcal.mol⁻¹, and E(tensor) <1400 kcal.mol⁻¹). For the F2-DNA and DNA-only ensembles, in order to sample conformation space adequately, it was found necessary to select the 1000 structures with the lowest XPLOR-NIH E(total) values from a total of 5000 calculated for each ensemble.

Finally, in the cases of the F1F2-DNA and F2-DNA ensembles, the additional atoms of the flexible N- and C-terminal protein tails and, in the case of the F1F2-DNA ensemble only, also the F1F2 linker (none of which were present in the protein template structures), were added

using a further simulated annealing protocol, during which the ordered portion of the structure (the whole DNA except for the 5′ phosphate group, as well as residues 6–91 of F1 and 109–201 of F2) was held rigid.

## Structural analysis

Our approach uses an ensemble of possible conformations of the macromolecular complex. For every structure, we compute the AVs of the donor and acceptor by using the structural analysis tool FastNPS[90]. As a user input, the algorithm needs the attachment point and the geometry of both the fluorophore and its linker, as well as the pdb-file of the structural ensemble. The dye and linker parameters were determined from the chemical structure using Chem3D 16.0 (PerkinElmer) and are summarized in Supplementary Table 4. With this information, FastNPS computes the volume which is accessible to the dye. These volumes are saved and used for the stochastic simulation of the FRET process.

As described in[45], it is assumed that the dyes can translate freely within the AVs. For rotation, we propose that the dyes diffuse in a spherical cone with a random axis obtained from the AV by a principle component analysis. As the diffusion coefficient of a dye relative to the molecule cannot be extracted from the smFRET measurements, a recently reported value of 10 Å²/ns is assumed for all dyes[42]. The semi-angle of the cone and the rotational diffusion coefficient are determined from time-resolved anisotropy data (Supplementary Table 5). For every conformation, we simulate 50,000 trajectories, i.e. the translational and rotational diffusion of donor and acceptor, and obtain an average smFRET efficiency. Using 50,000 trajectories per structure reduced the standard deviation of 100 independent simulations for the same model to 0.2% (as compared to 0.5% for 10,000 or 1.7% for 1000 simulated excitations), so that the statistical error is small compared to the overall uncertainty of the simulations. The parameters used in the simulations are given in Supplementary Table 4 and Supplementary Table 5. Details about their determination are given in the section "Data analysis" and "Determination of the isotropic Förster radius" (Supplementary methods).

We select conformations whose simulated efficiencies are within an error of $\Delta E = 0.05$ compared to the experimental result. This accounts for both the error in the isotropic Förster Radius ($\Delta R_{iso} = 2$ Å translates to $\Delta E = 0.04$ in the simulations, see also section "Determination of the isotropic Förster radius") and the error of the measurement derived from the uncertainties in the correction factors ($\Delta E = 0.03$, see section "Burst selection and quantification of burst-wise parameters").

Every linear dsDNA molecule has a unique axis. We are interested in the angle between the axes of the 5′- and 3′-stems. As a convention, the direction of their axes is defined from the nick to the respective loop. We define the axis of each DNA stem as the longest axis determined from a principal component analysis, considering all atoms belonging to the respective stem, but omitting the four bases of the tetraloop and the base pair next to the loop. All structures are aligned with respect to the 3′-stem, such that we can represent every selected conformation as a unit vector/orientation (representing the 5′-stem) in a spherical coordinate system. This allows us to compute the angle between the axes of the two stems for every conformation. We obtain histograms of these angles where we re-weight the relative frequency in the equally spaced bins according to the differing surface of the ring on the unit sphere. For the computation of mean angles and standard deviations, we considered two axially symmetric scenarios: the orientations lie in a spherical cone or in an open spherical sector (the symmetric difference of two cones with the same mean axis). In the view along the mean axis, they represent a circle and an annulus, respectively (Supplementary Fig. 21). In the case of a spherical cone, we have a single mean axis which is

computed by the normalized sum over all axes. The standard deviation is retrieved from the corrected histogram. If the orientations lie within an open spherical sector, we do not have one unique mean orientation, but it is axially degenerate, i.e. it is on a lateral surface of a cone. However, the mean polar angle between the stems is still uniquely defined, if we assume that the open spherical sector is directed along the 3'-axis. Thus, both the mean angle and the standard deviation are computed by the corrected histogram.

Pictures of molecular structures were generated using the UCSF Chimera package[91]. Chimera is developed by the Resource for Bio-computing, Visualization, and Informatics at the University of California, San Francisco (supported by NIGMS P41-GM103311).

### Filtered FCS
Filtered FCS (fFCS) was employed to disentangle the mixture of species and to filter out the separate species using their difference in FRET efficiency[56]. Thus, to separate the interconverting states in DNA with F2, DNA with F1F2 and DNA with PARP-1 datasets, burstwise fFCS with a time window of 50 ms around the edges of each burst was performed using the MATLAB package PAM[82] as described in[57]. The bursts corresponding to separate species (states) were selected by setting the respective FRET efficiency thresholds (species 1: $0.25 < E < 0.35$, species 2: $0.77 < E < 0.87$ for DNA+F2; species 1: $0.34 < E < 0.42$, species 2: $0.9 < E < 0.98$ for DNA with either F1F2, PARP-1, or PARP-1 in presence of PARPi). The FRET efficiency thresholds were chosen to be close to the edges of FRET histograms in order to avoid mixture of the two interconverting species. For every species, TCSPC patterns in donor and FRET channels were plotted from which the stacked photon histograms - fFCS filters were generated. The filters of each species were used to calculate the auto- and cross-correlation functions[56]. For each dataset (Figs. 5, 6 and Supplementary Fig. 10) four correlation functions were calculated, namely species 1 x species 1, species 1 x species 2, species 2 x species 1, species 2 x species 2. The correlation functions were fit globally with a diffusion ($G_{diff}$) model containing two kinetic terms ($\tau_{R,i}$):

$$G(\tau) = G_{diff}(\tau)\left[1 + \sum_{i=1}^{2} A_i \exp\left(-\frac{\tau}{\tau_{R,i}}\right)\right] + offset$$

$$G_{diff}(\tau) = \frac{1}{N\sqrt{8}}\left[1 + \frac{\tau}{\tau_D}\right]^{-1}\left[1 + \frac{\tau}{p^2\tau_D}\right]^{-1/2}$$

Again, N is the average number of the observed molecules in the effective observation volume, $\tau_D$ is the translational diffusion time, $p$ is the ratio of the lateral and axial dimensions of the confocal volume. The diffusion time $\tau_D$ and one relaxation time $\tau_{phot}$ were fixed to the values obtained by FRET-FCS (Supplementary Methods) and the other relaxation time $\tau_R$ was globally linked for the four correlation functions. The amplitudes of the species cross-correlation functions were allowed to be negative. Standard error of the mean (SEM, Table 1) was calculated by splitting the respective dataset into 5 to 8 equal time periods. Every time period was subjected to fFCS analysis and the resulting interconversion times were then used for SEM calculation.

### Reporting summary
Further information on research design is available in the Nature Research Reporting Summary linked to this article.

## Data availability
The smFRET data generated in this study have been deposited in the Dryad Digital Repository database under accession code dryad.66t1g1k3v

## Code availability
DNA kink angle analysis code is available online through the GitHub Repository (https://github.com/Michaelislab/-DNA-kink-angle-from-smFRET-analysis).

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

## Acknowledgements

We are grateful for the support from Nadine Jakobi, performing the labeling of the DNA donor strand, Mara Guriento who performed F2 purification, Laura Easton, who provided the protocol for PARP-1 purification in *E.coli*, Katharina Werner and Camilla Förster, who helped with smFRET measurements.This work was supported by the Deutsche Forschungsgemeinschaft through the CRC 1279 (JM). DN was supported by the Medical Research Council [U105178934]. EK acknowledges support by the Carl-Zeiss Stiftung.

## Author contributions

J.M. and S.E. designed the study. A.S. and E.K. performed the experiments and analysed the data. T.E developed the angle quantification analysis. C.R. provided technical support for the experimental setup for the smFRET analysis. D.N generated all the structural ensembles and provided the plasmid DNA for PARP-1. A.S. O.K. and S.E. expressed and purified proteins. E.K., A.S., J.M., D.N., O.K., and S.E. wrote the manuscript.

## Funding

## Competing interests

The authors declare no competing interests.
