## [Peer Review File · Nature Communications]

Structural dynamics of DNA strand break sensing by PARP-1 at a single-molecule levelREVIEWER COMMENTS

Reviewer #1 (Remarks to the Author):

In this manuscript Sefer, Kallis et al examine the dynamics of PARP1 binding to an SSB using smFRET and structural modelling. PARP1 binding to an SSB was shown to induce a cascade of conformational changes spreading from the N-terminal zinc fingers to the WGR, helical and catalytic domain to activate PARP1. To address what happens with the conformation of the DNA substrate, the authors used an SSB-containing DNA substrate labelled with two fluorophores on either side of the nick to examine structural changes upon PARP1 binding to nicked DNA based on FRET between the two fluorophores. The authors found that nicked DNA has the same conformation as intact DNA but undergoes kinking upon PARP1 binding, in accordance with an induced fit model. Furthermore, using structural modelling and computational analysis the authors determined the conformations that correspond to measured FRET efficiencies and used those to calculate mean kinking angles. The kinking is initiated through the F2 zinc finger and enhanced upon subsequent binding of the F1 zinc finger. PARP inhibitors show differential effects on DNA kinking. The data presented in this manuscript are robust and the conclusions are sound and justified. I am missing, however, some insights into the functional relevance of PARP1-induced DNA kinking.

Major point:

1) Why does PARP1 induce a 90° kink in nicked DNA? F2-induced kinking facilitates F1 binding to the 5' stem but why is DNA further twisted upon F1 binding? Does this facilitate assembly of SSB factors? Could the authors test (i) whether the addition of XRCC1, Pol beta, FEN1 affects DNA kinking and (ii) how the presence of F2 vs F1F2 (different degrees of DNA kinking) affect binding affinity between those SSB factors and nicked DNA?

Minor suggestions:

1) Abstract: I would remove the following statement: 'the data is inconsistent with a conformational selection model' for two reasons. Firstly, this term is too specialized and would need to be explained first. Secondly, this manuscript does not focus on refuting an existing concept but revealing a new finding.

2) Fig. 6b – please use more distinct colours

Typos:

1) line 87, 467: adopts, adopted

2) Quantification of kink angles has a different formatting compared to other subsections.

3) line 387: gapped

4) line 455: F2 binding to the 3' stem

Reviewer #2 (Remarks to the Author):

Key Results

Understanding the mechanisms of DNA damage sensing by PARP-1 is important as it plays a key role in several DNA repair pathways. In addition, as a frequent chemotherapy target using the principle of synthetic lethality, PARP-1 has clinical relevance. As the authors point out, understanding the roles of both protein and DNA conformational changes in damage recognition is necessary for a complete mechanism.

The authors employ sophisticated smFRET and FCS experimental measurements on diffusing double-labeled nicked DNA substrates in the presence of full length and truncated PARP-1. In doing so, they shed light on the individual roles of two zinc finger domains in kinking the DNA for damage recognition and downstream processing. In combination with computational modeling, the authors identify reasonable DNA structures and extract bend angles that agree with experimental smFRET data.

Further data analysis reveals a highly dynamic PARP1-nicked DNA complex. Addition of PARP-1 inhibitors stabilizes either high or low FRET states providing key insights into the impact of inhibitors on molecular function.

Validity

A key claim of this paper is that the data taken together suggest an induced fit mechanism as opposed to conformational selection, which is driven by a multidomain assembly cascade of the enzyme. The F2 fragment of PARP-1 is enough to kink the DNA a little bit, and the F1F2 fragment of PARP-1 is enough to highly kink the DNA. The data suggest that initial binding of F2 kinks the DNA a little bit so that F1 can bind, then F1 binds and kinks the DNA to a higher degree. This suggested cascade of events may then lead to the activation of PARP-1 catalytic activity. Although it is not a direct observation of protein domain assembly, advanced analysis of TCSPC data (FRET-FCS and fFCS) lends strong support to this claim. The results reveal a highly dynamic protein-DNA complex with a structurally distinct F2-DNA kinked state.

Data and methodology

The authors have utilized TCSPC that is spectrally and polarization-resolved in combination with pulsed interleaved excitation, leading to a wealth of high temporal resolution data including smFRET, FCS, fluorescence lifetimes, and anisotropy. Monitoring diffusing molecules (as opposed to immobilized molecules) avoids surface interactions and artifacts, although it may be difficult to precisely determine a sequence of events due to diffusion. Extracting distance information from smFRET must be carefully done, but the authors' hybrid experimental - computational approach is novel and can be broadly applied. Additional experiments in the presence of inhibitors are a nice touch that emphasize the clinical relevance of studying these interactions.

Analytical Approach

The authors exploit the full power of TCSPC data using E vs. τ D plots, BVA, and time-window analysis to confirm dynamics of the protein-DNA complex. Additional analysis of TCSPC data by FRET-FCS and fFCS further strengthens the results. Analyzing the different species revealed in FRET efficiency histograms, they can distinguish between structural heterogeneity and fast conformational changes.

Additional notes:

- In all experiments, it seems that appropriate controls have been performed. Repeating the experiments with a different donor helps to rule out photophysical artifacts.
- It appears that all the fluorescence signals have been appropriately corrected.
- The hybrid analytical approach is broadly useful and accessible to the smFRET community, especially with the use of open-source analysis software (i.e. PAM).
- The spherical coordinate system representation of bend angle is helpful.
- The authors have been rigorous about error propagation.

Suggested improvements

- Some mention in methods or SI about why such a large molar excess of protein (compared to DNA concentration) is used. Is the large molar excess necessary to see the DNA kinking effect or is there some other factor?
- Sample size – it may be useful to report a usual data acquisition time and average number of molecules that diffuse through the confocal volume during that time so that readers understand sample size in this type of experiment.
- Can you comment on why a dumbbell structure was chosen as opposed to a dsDNA construct?

- Perhaps add molecular weight of fragments and fl PARP-1 to Supp. Table S3

Clarity and context

Beginning on line 46, "However, as yet no mechanistic insight was obtained...", it is unclear which protein-DNA complex is being referred to. Earlier in that paragraph, the authors cite several relevant examples of smFRET experiments on DNA kinking/bending dynamics for different protein-DNA complexes. For the MutS-DNA example (Sass et al. 2010), the authors are referred to a follow up article (LeBlanc et al. 2018, doi: 10.1093/nar/gky865) that uses two labeling strategies in smFRET experiments to reveal the pathway of coordinated conformational changes in Taq MutS and mismatch DNA that are important for mismatch recognition. Perhaps that citation is relevant there or on line 447 with Lamers et al. 2000.

Line 118 – It may be helpful to the reader to explicitly state number of bases between dye attachment sites here

Line 139 - It is unclear here how the FRET efficiency error is calculated

Line 165 – It may be useful to the reader to note that the experimental observation time of 1 ms is due to binning, presumably

Line 295 – "only very small changes were observed..." is a bit vague, may be unclear to reader what small changes are being referred to

Lines 309-316 Seems a bit out of place. If left here, the authors should at least point the reader to the FRET efficiency histograms that the FRET-FCS data are extracted from (showing the location of the low FRET populations being referred to). The niraparib data in Fig.6 is also used for this analysis, but has not been introduced in the main text yet, so it is a bit confusing.

Line 320 – It may not be obvious to the reader why you selected efficiency regions toward the edge of the histogram or how your FRET efficiency histogram thresholds were chosen. Perhaps a bit more clarity about that in Methods would be helpful.

Figure 5 – I believe these are the same histograms from Figure 2. That is not clear from the caption.

Figure 6 – Data from 'DNA with EB-47' is omitted and it is not clear why when 'DNA with niraparib' is included in 6b. Perhaps also label the E(LF) population mentioned in line 349 on the figure.

Line 415 – It may be useful to the reader to include the average bend angles again here in parentheses for free DNA (...) and DNA+F1F2 (...)

Overall: Very nice, well-written paper, interesting work and analysis.

Reviewer #3 (Remarks to the Author):

Structural dynamics of DNA strand break sensing by PARP-1 at a single-molecule level

Sefer et al develop a FRET-based assay for measuring PARP1 impact on the structure of single-strand break DNA, using a DNA dumbbell model. The experiments are well described and well done and the manuscript is easy to read and straightforward. The results are consistent with published reports.

However, the manuscript falls short of providing major new insights into PARP1 interaction/dynamics with DNA single-strand breaks. Most of the manuscript is spent describing the methodology, rather than using the new assay to the fullest extent. For example, only a few fragments of PARP1 are tested in the study, and they seem to be the minimal set to support the model they propose. What is the effect of F1 on the DNA structure? What are the effects of mutations to F2 that inactivate DNA binding? F2 is proposed to play a central role in the proposed dynamic model, so it is important to test the loss of F2 function (through deletion or mutagenesis). There is also an opportunity to test PARP1 mutations in other domains, for example the WGR that is involved in the "monkey bar" mechanism and PARP1 allosteric signalling. These experiments could better capture the usefulness of the approach, perhaps providing new insights into deficiencies of mutants. Moreover, these experiments are important for supporting the model that is proposed where F2 is the key initiator of DNA binding. As another example, only two PARP inhibitor compounds are tested. A larger panel of PARP inhibitors could indicate whether the FRET-based assay will be able to discern subtle differences between inhibitors and their impact on PARP1 interaction with DNA. There are other questions that could be tested in the assay, such as the effect of adding NAD⁺, or the effect of changes to the DNA structure (size of the gap at the nick), or the impact of the key accessory factor HPF1. Without extending the assay into these other areas, or more thoroughly testing the proposed model with additional PARP1 fragments/mutants, the study seems largely technical without pushing the boundaries of understanding of PARP1.

Other points:

The Le Cam (1994) reference should be noted earlier in the study (currently not until the Discussion), as the DNA bending angle was analyzed in this work.

Second Discussion paragraph, line 378, gaped is used instead of gapped.

The change in buffer conditions between F1F2/F2 versus PARP1 should be explained. Why was this change needed? Ideally the buffers would maintain the same ionic strength.

It seems that only one protein concentration is reported for each protein. Were others tested? What is the protein concentration dependence of the FRET signal?

We wish to thank all three reviewers for the in-depth analysis of our manuscript. In response to the suggestions we have performed numerous additional experiments which we are happy to report in the revised version of the manuscript. Below you will find a detailed response to all.

Reviewer #1 (Remarks to the Author):

In this manuscript Sefer, Kallis et al examine the dynamics of PARP1 binding to an SSB using smFRET and structural modelling. PARP1 binding to an SSB was shown to induce a cascade of conformational changes spreading from the N-terminal zinc fingers to the WGR, helical and catalytic domain to activate PARP1. To address what happens with the conformation of the DNA substrate, the authors used an SSB-containing DNA substrate labelled with two fluorophores on either side of the nick to examine structural changes upon PARP1 binding to nicked DNA based on FRET between the two fluorophores. The authors found that nicked DNA has the same conformation as intact DNA but undergoes kinking upon PARP1 binding, in accordance with an induced fit model. Furthermore, using structural modelling and computational analysis the authors determined the conformations that correspond to measured FRET efficiencies and used those to calculate mean kinking angles. The kinking is initiated through the F2 zinc finger and enhanced upon subsequent binding of the F1 zinc finger. PARP inhibitors show differential effects on DNA kinking. The data presented in this manuscript are robust and the conclusions are sound and justified.

We wish to thank the reviewer for the positive assessment of the paper.

I am missing, however, some insights into the functional relevance of PARP1-induced DNA kinking.

Major point:

1) Why does PARP1 induce a 90° kink in nicked DNA? F2-induced kinking facilitates F1 binding to the 5' stem but why is DNA further twisted upon F1 binding? Does this facilitate assembly of SSB factors? Could the authors test (i) whether the addition of XRCC1, Pol beta, FEN1 affects DNA kinking and (ii) how the presence of F2 vs F1F2 (different degrees of DNA kinking) affect binding affinity between those SSB factors and nicked DNA?

We agree with the reviewer that our experiments and results would be nicely complemented with other functional data. As suggested by the reviewer we performed additional experiments to test whether PARP-1 induced kinking is altered by the presence of XRCC1 and whether the presence of F1F2 or F2 affect XRCC1 binding. The results of these experiments are presented in the new Fig.7 and new supplemental figures Supp. Fig.S11-S14.

Moreover, we present a structural analysis of Fen1 and Pol beta binding to the DNA ligand based on previously published data and compare it to the binding of F1F2. We find that the binding sites overlap substantially as shown in supplementary figure Supp. Fig. S17, preventing these proteins from binding at the same time. It would be interesting to study at a single molecule level how XRCC-1 coordinates more downstream hand-off events, but this would be beyond the scope of the current paper.

Minor suggestions:

1) Abstract: I would remove the following statement: 'the data is inconsistent with a

conformational selection model' for two reasons. Firstly, this term is too specialized and would need to be explained first. Secondly, this manuscript does not focus on refuting an existing concept but revealing a new finding.

We adapted the abstract in the revised version and no longer mention the term “conformational selection”.

2) Fig. 6b – please use more distinct colours

We have adjusted the colors.

Typos:

1) line 87, 467: adopts, adopted

2) Quantification of kink angles has a different formatting compared to other subsections.

3) line 387: gapped

4) line 455: F2 binding to the 3' stem

We have corrected the typos and formatting.

Reviewer #2 (Remarks to the Author):

Key Results

Understanding the mechanisms of DNA damage sensing by PARP-1 is important as it plays a key role in several DNA repair pathways. In addition, as a frequent chemotherapy target using the principle of synthetic lethality, PARP-1 has clinical relevance. As the authors point out, understanding the roles of both protein and DNA conformational changes in damage recognition is necessary for a complete mechanism.

The authors employ sophisticated smFRET and FCS experimental measurements on diffusing double-labeled nicked DNA substrates in the presence of full length and truncated PARP-1. In doing so, they shed light on the individual roles of two zinc finger domains in kinking the DNA for damage recognition and downstream processing. In combination with computational modeling, the authors identify reasonable DNA structures and extract bend angles that agree with experimental smFRET data. Further data analysis reveals a highly dynamic PARP1-nicked DNA complex. Addition of PARP-1 inhibitors stabilizes either high or low FRET states providing key insights into the impact of inhibitors on molecular function.

Validity

A key claim of this paper is that the data taken together suggest an induced fit mechanism as opposed to conformational selection, which is driven by a multidomain assembly cascade of the enzyme. The F2 fragment of PARP-1 is enough to kink the DNA a little bit, and the F1F2 fragment of PARP-1 is enough to highly kink the DNA. The data suggest that initial binding of F2 kinks the DNA a little bit so that F1 can bind, then F1 binds and kinks the DNA to a higher degree. This suggested cascade of events may then lead to the activation of PARP-1 catalytic activity. Although it is not a direct observation of protein domain assembly, advanced analysis of TCSPC data (FRET-FCS and fFCS) lends strong support to this claim. The results reveal a highly dynamic protein-DNA complex with a structurally distinct F2-DNA kinked state.

Data and methodology

The authors have utilized TCSPC that is spectrally and polarization-resolved in combination with pulsed interleaved excitation, leading to a wealth of high temporal resolution data including smFRET, FCS, fluorescence lifetimes, and anisotropy. Monitoring diffusing molecules (as opposed to immobilized molecules) avoids surface interactions and artifacts, although it may be difficult to precisely determine a sequence of events due to diffusion. Extracting distance information from smFRET must be carefully done, but the authors' hybrid experimental - computational approach is novel and can be broadly applied. Additional experiments in the presence of inhibitors are a nice touch that emphasize the clinical relevance of studying these interactions.

Analytical Approach

The authors exploit the full power of TCSPC data using E vs. τ_D plots, BVA, and time-window analysis to confirm dynamics of the protein-DNA complex. Additional analysis of TCSPC data by FRET-FCS and fFCS further strengthens the results. Analyzing the different species revealed in FRET efficiency histograms, they can distinguish between structural

heterogeneity and fast conformational changes.

Additional notes:

- In all experiments, it seems that appropriate controls have been performed. Repeating the experiments with a different donor helps to rule out photophysical artifacts.
- It appears that all the fluorescence signals have been appropriately corrected.
- The hybrid analytical approach is broadly useful and accessible to the smFRET community, especially with the use of open-source analysis software (i.e. PAM).
- The spherical coordinate system representation of bend angle is helpful.
- The authors have been rigorous about error propagation.

We are very grateful to the reviewer for the positive assessment of our paper.

Suggested improvements

- Some mention in methods or SI about why such a large molar excess of protein (compared to DNA concentration) is used. Is the large molar excess necessary to see the DNA kinking effect or is there some other factor?

We thank the reviewer for bringing this up. In the revised version of the manuscript we are now showing titrations of protein concentrations in the new Supp. Fig. S18. We have added a sentence to the methods section to clarify our choice of concentrations (page 29, lines 723-724).

- Sample size – it may be useful to report a usual data acquisition time and average number of molecules that diffuse through the confocal volume during that time so that readers understand sample size in this type of experiment.

We agree with the reviewer that this information is useful. In the revised version of the manuscript, we go even a step further and provide a new table (Supp. Table S8), giving detailed information about the statistics for each experiment performed.

- Can you comment on why a dumbbell structure was chosen as opposed to a dsDNA construct?

PARP-1 also recognizes blunt ends. Therefore, in order to maximize the specificity of binding to the designed lesion in the DNA, we have used the tetraloops, to minimize binding to the ends of the stems. This strategy has been established for PARP-like zinc fingers in previous studies (PMID: 15288782, PMID: 21262234) and enabled key subsequent studies (PMID 32241924). We have added a sentence to the revised manuscript clarifying this point (page 26, lines 637-638)

- Perhaps add molecular weight of fragments and fl PARP-1 to Supp. Table S3

We have added molecular weights for all proteins used in this study to Supp. Table S3.

Clarity and context

Beginning on line 46, “However, as yet no mechanistic insight was obtained...”, it is unclear which protein-DNA complex is being referred to. Earlier in that paragraph, the authors cite several relevant examples of smFRET experiments on DNA kinking/bending dynamics for

different protein-DNA complexes. For the MutS-DNA example (Sass et al. 2010), the authors are referred to a follow up article (LeBlanc et al. 2018, doi: 10.1093/nar/gky865) that uses two labeling strategies in smFRET experiments to reveal the pathway of coordinated conformational changes in Taq MutS and mismatch DNA that are important for mismatch recognition. Perhaps that citation is relevant there or on line 447 with Lamers et al. 2000.

We have realized that this statement was too general and have therefore changed the sentence (page 1, line 46). We have also added the suggested reference (page 1, line 40).

Line 118 – It may be helpful to the reader to explicitly state number of bases between dye attachment sites here

We have added the number of bases between attachment sites to the results section of the manuscript (Page 5, line 140)

Line 139 - It is unclear here how the FRET efficiency error is calculated

In the revised manuscript at this point (page 6, line 161) we are referring to the methods section where we explicitly explain how the error is calculated (Equations 1 and 1a-1f)

Line 165 – It may be useful to the reader to note that the experimental observation time of 1 ms is due to binning, presumably

We have added “binning time” to the mentioned statement.

Line 295 – “only very small changes were observed...” is a bit vague, may be unclear to reader what small changes are being referred to

We have clarified our statement by writing “... the resulting histograms at shorter time bins become skewed, but no distinct sub-populations become visible, indicating that dynamics occur on an even faster time-scale.”

Lines 309-316 Seems a bit out of place. If left here, the authors should at least point the reader to the FRET efficiency histograms that the FRET-FCS data are extracted from (showing the location of the low FRET populations being referred to). The niraparib data in Fig.6 is also used for this analysis, but has not been introduced in the main text yet, so it is a bit confusing.

We agree, that the paragraph was somewhat confusing. We have modified the beginning of the paragraph to better explain why we were performing the additional FRET-FCS analysis (page 14, line 325). Moreover, we also added the reference to the original figure to the text, as suggested by the reviewer. Lastly, we have moved the FRET-FCS discussion of the data in presence of the PARPi to a later section of the paper.

Line 320 – It may not be obvious to the reader why you selected efficiency regions toward the edge of the histogram or how your FRET efficiency histogram thresholds were chosen. Perhaps a bit more clarity about that in Methods would be helpful.

We have extended the discussion of the fFCS approach in the Methods section, explaining the choice of the chosen FRET efficiency regions, by adding: The FRET efficiency thresholds were chosen to be close to the edges of FRET histograms in order to avoid mixture

of the two interconverting species.

Figure 5 – I believe these are the same histograms from Figure 2. That is not clear from the caption.

It is correct that these are the same data and are just shown in both locations to facilitate comparison. We have explicitly mentioned this in the revised Figure caption of Figure 5 (and elsewhere when appropriate).

Figure 6 – Data from ‘DNA with EB-47’ is omitted and it is not clear why when ‘DNA with niraparib’ is included in 6b. Perhaps also label the E(LF) population mentioned in line 349 on the figure.

We had previously only shown DNA with niraparib rather than pure DNA because, here, the observed photo-physics caused a change in the determined smFRET efficiency. In the revised version of the paper we have added Supp. Fig. S10 showing the changes to the observed smFRET histogram for DNA for all inhibitors.

Line 415 – It may be useful to the reader to include the average bend angles again here in parentheses for free DNA (...) and DNA+F1F2 (...)

We have added the angles at the mentioned position.

Overall: Very nice, well-written paper, interesting work and analysis.

Reviewer #3 (Remarks to the Author):

Structural dynamics of DNA strand break sensing by PARP-1 at a single-molecule level

Sefer et al develop a FRET-based assay for measuring PARP1 impact on the structure of single-strand break DNA, using a DNA dumbbell model. The experiments are well described and well done and the manuscript is easy to read and straightforward.

We wish to thank reviewer 3 for the positive statement regarding our experiments and writing.

The results are consistent with published reports. However, the manuscript falls short of providing major new insights into PARP1 interaction/dynamics with DNA single-strand breaks. Most of the manuscript is spent describing the methodology, rather than using the new assay to the fullest extent. For example, only a few fragments of PARP1 are tested in the study, and they seem to be the minimal set to support the model they propose. What is the effect of F1 on the DNA structure? What are the effects of mutations to F2 that inactivate DNA binding? F2 is proposed to play a central role in the proposed dynamic model, so it is important to test the loss of F2 function (through deletion or mutagenesis). There is also an opportunity to test PARP1 mutations in other domains, for example the WGR that is involved in the "monkey bar" mechanism and PARP1 allosteric signalling. These experiments could better capture the usefulness of the approach, perhaps providing new insights into deficiencies of mutants. Moreover, these experiments are important for supporting the model that is proposed where F2 is the key initiator of DNA binding. As another example, only two PARP inhibitor compounds are tested. A larger panel of PARP inhibitors could indicate whether the FRET-based assay will be able to discern subtle differences between inhibitors and their impact on PARP1 interaction with DNA. There are other questions that could be tested in the assay, such as the effect of adding NAD⁺, or the effect of changes to the DNA structure (size of the gap at the nick), or the impact of the key accessory factor HPF1. Without extending the assay into these other areas, or more thoroughly testing the proposed model with additional PARP1 fragments/mutants, the study seems largely technical without pushing the boundaries of understanding of PARP1.

We would like to thank the reviewer for the in-depth discussion, which stimulated several new experiments as discussed in the revised manuscript. First, we use the concrete suggestion about PARP inhibitors and included four additional inhibitors for a total of 6 different inhibitors, covering all three proposed classes of inhibitors (Figure 7 and Supp. Fig. S9). We also show in the revised manuscript that our methodology is in fact able to detect subtle changes between the different inhibitors, which are best described by the differences in the observed relaxation rates between the kinked and linear conformation (Table 1). We have also expanded our discussion by comparing to a recent publication from the laboratory of Karolin Luger (p. 23, lines 570-573, PMID: 35259019).

Moreover, we would like to state, that the experiments performed in response to the comments from reviewer 1, namely studying the binding of XRCC1 in presence and absence of F2 and F1F2 as well as the kinking changes in presence of XRCC1 (compare to new Figure 7 and Supp. Fig. S12), while not directly mentioned by the reviewer, address the mentioned concern regarding the application of the assay to a biological problem. These new experiments certainly

provide new insight into PARP1 DNA recognition and the integrative function of XRCC1. While we also believe that experiments probing the previously described “monkey bar” mechanism in context of PARP-1 mutants and XRCC1 could yield important novel insight, these are beyond the scope of the present work, and we believe that the provided new data adequately addresses the concerns of the reviewer.

Other points:

The Le Cam (1994) reference should be noted earlier in the study (currently not until the Discussion), as the DNA bending angle was analyzed in this work.

We have now mentioned this reference already in the introduction of the paper (page 3 Line 90).

Second Discussion paragraph, line 378, gaped is used instead of gapped.

We have corrected this typo.

The change in buffer conditions between F1F2/F2 versus PARP1 should be explained. Why was this change needed? Ideally the buffers would maintain the same ionic strength.

In the revised version of the manuscript, we present additional data (Supp. Fig. S19) to explain our choice of buffer conditions.

It seems that only one protein concentration is reported for each protein. Were others tested? What is the protein concentration dependence of the FRET signal?

In the revised version of the manuscript we are now presenting additional data showing how differences in the protein concentration influence the observed smFRET distributions (Supp. Fig. S18).

REVIEWER COMMENTS

Reviewer #1 (Remarks to the Author):

The authors have addressed all my concerns and thus I fully support publication of this manuscript.

Reviewer #2 (Remarks to the Author):

The authors have satisfactorily addressed my questions and comments in the revised manuscript. Thank you.

Reviewer #3 (Remarks to the Author):

The re-submitted manuscript addresses some of the issues raised during the initial review; however, some concerns remain and new concerns are raised. The work remains largely a technical advance in following DNA break structure in the presence of PARP1. I think the suggestions below could help the presentation of the findings such that the potential implications are clearer to readers.

I do not feel like the revised version answers the question of why F2 is used at such high concentrations. A supplemental figure shows that the observed changes in DNA structure with F2 alone require >4 μM concentration, which indicates that F2 alone is unlikely to kink the DNA to an appreciable extent without the aid of other PARP1 domains. What happens when F1 is added at 10, 20, 40 μM concentrations? Statements like this "our data shows that DNA kinking induced by F2 enables not only binding of F1 and the subsequent domain assembly cascade leading to allosteric activation of PARP-1, but stimulates also the interplay with XRCC-1." are not really consistent with the quite weak DNA kinking activity of F2 alone. It is not clear to me why F2 has to be presented as the initiating driving force. It clearly doesn't act alone.

Regarding the new data for XRCC1, it is hard to decipher what binding mode(s) is/are expected for XRCC1. It would appear that XRCC1 might favor the kinked conformation of DNA that PARP1 induces, but shouldn't that conformation be evident at least at some level in the experiments with only DNA and XRCC1? Or does XRCC1 have a different mode of DNA binding in the absence versus presence of PARP1? Is there a direct PARP1-XRCC1 interaction that would support a different binding mode for XRCC1?

What are the errors associated with the diffusion times reported in Figure 7?

I could not find the concentrations used for the different PARPi. Is this listed somewhere? How do the concentrations compare to reported binding affinities or IC50 values? Was the effect of the PARPi concentration dependent?

It seems that the analysis of PARPi effects on the DNA state distribution and/or interconversion rates is consistent with the classification of PARPi based on allosteric effects on DNA binding and hydrogen exchange, but it is not clear how separated they are. In some cases, the smFRET efficiency histograms show fairly clear evidence of a shift in populations (EB-47 and niraparib), and the interconversion times in Table 1 are also fairly distinct (although it would be useful to have some statistical analysis of the differences). For the other cases (talazoparib, veliparib, rucaparib, olaparib), the smFRET efficiency histograms do not appear to show major differences, but the interconversion times are distinct (statistical analysis of the differences would also help here). Is there a difference between niraparib/rucaparib versus veliparib that can be inferred from these differences?

"In this structure, the gapped DNA adopts a highly kinked conformation, similar to what had been seen for nicked DNA in early electron microscopy studies (Le Cam et al. 1994), F2 is bound to the side of the gap bearing the 3' terminus, F1 binds to the side bearing the 5' terminus, and the two zinc fingers form interdomain contacts with one another."

I suggest breaking this sentence in two, by placing a period after the Le Cam reference.

"which might help to explain the different reported potencies of these inhibitors in vitro and in vivo (Rudolph, Jung, and Luger 2022)."

It seems appropriate to cite the primary literature reporting these differences.

The below reference was listed twice (and has a formatting issue).

Langelier, Marie-france, Poly Adp-ribosyl, Jamie L Planck, Swati Roy, and John M Pascal. 2012.

"Structural Basis for DNA Damage-Dependent Poly(ADP-Ribosylation) by Human PARP-1" 336 (6082): 728-32. 1032 <https://doi.org/10.1126/science.1216338>.

The publication PMID: 21233213 really should be referenced in this study, as structures from this publication (3oda, 3odc) are specifically cited in the methods.

"Yet, we find that the diffusion time determined by FCS shows an increase in presence of XRCC1 (Supp. Fig. S12a lower panel)"

Seems that this should be Fig. S12c.

"Interestingly, class III PARP-1 inhibitors such as niraparib have the opposite effect, the kinked state is de-stabilised as predicted by the hydrogen exchange data (Zandarashvili et al. 2020) and the linear conformation becomes more prominent (Fig. 6b)."

Shouldn't this be Figure 6c?

We wish to thank Reviewer #1 and #2 for their appreciation of our work.

Reviewer #3 (Remarks to the Author):

The re-submitted manuscript addresses some of the issues raised during the initial review; however, some concerns remain and new concerns are raised. The work remains largely a technical advance in following DNA break structure in the presence of PARP1. I think the suggestions below could help the presentation of the findings such that the potential implications are clearer to readers.

We wish to thank reviewer #3 for the suggestions for making the implications of our work clearer to the readers.

I do not feel like the revised version answers the question of why F2 is used at such high concentrations. A supplemental figure shows that the observed changes in DNA structure with F2 alone require >4 μM concentration, which indicates that F2 alone is unlikely to kink the DNA to an appreciable extent without the aid of other PARP1 domains. What happens when F1 is added at 10, 20, 40 μM concentrations? Statements like this "our data shows that DNA kinking induced by F2 enables not only binding of F1 and the subsequent domain assembly cascade leading to allosteric activation of PARP-1, but stimulates also the interplay with XRCC-1." are not really consistent with the quite weak DNA kinking activity of F2 alone. It is not clear to me why F2 has to be presented as the initiating driving force. It clearly doesn't act alone.

The reviewer is correct that F2 doesn't act alone and that we are using concentrations well above of the published dissociation constants, even for individual fingers F1 and F2. We acknowledge that these points were perhaps not clear enough in our manuscript. We have revised our discussion accordingly. As suggested by the reviewer, we provide now also additional smFRET data for direct comparison between F1 and F2. These experiments help indeed to further clarify the mechanistic role of F1 and F2.

While F1 binding to DNA and DNA nicks has been reported to be less strong than that of F2 (compare Langelier 2011 and Eustermann 2011), a lengthening of the diffusion time in comparison to the free DNA directly shows that F1 binds DNA at a concentration of $10\mu\text{M}$. In contrast to F2, however, F1 does not alter the conformation of DNA, it stays linear. These data are also in agreement with the observed in-vivo difference in recruitment to the site of DNA lesions. While F2 recruitment is weak in comparison to F1F2 or full-length PARP-1 in life cell experiments (yes, it doesn't work alone), F1 recruitment cannot be detected (compare Figure 1A Eustermann et al. Mol. Cell 2015). Both F2 and F1 need to cooperate together in DNA damage recognition (compare also to the mutational analysis Ali et al. NSMB 2012). Both F2 and F1 work together in the error recognition and we have clarified these points in the results and discussion section of the revised manuscript. We have removed the mentioned statement about XRCC-1 from the results section.

Regarding the new data for XRCC1, it is hard to decipher what binding mode(s) is/are expected for XRCC1. It would appear that XRCC1 might favor the kinked conformation of DNA that PARP1 induces, but shouldn't that conformation be evident at least at some level in the experiments with only DNA and XRCC1? Or does XRCC1 have a different mode of DNA binding in the absence versus presence of PARP1? Is there a direct PARP1-XRCC1 interaction that would support a different binding mode for XRCC1?

Our data clearly shows that binding of XRCC1 to DNA does not lead to a detectable conformational change. In contrast presence of F1F and F2 helps XRCC1 binding through a direct protein-protein interaction or indirectly by recognizing the kinked DNA state. We have clarified this point in the discussion section. While a potential binding mode to the kinked DNA damage site has been proposed in the literature, we believe that binding to DNA in absence of F2 or F1F2 is non-specific. We have clarified these points in the updated manuscript and have added corresponding cartoons to the figures to help the interpretation of the data.

What are the errors associated with the diffusion times reported in Figure 7?

We thank the reviewer for bringing up this important point in the data analysis. We have now added error bars to all reported diffusion times and added a statement on how the errors were determined to the Methods section.

I could not find the concentrations used for the different PARPi. Is this listed somewhere? How do the concentrations compare to reported binding affinities or IC50 values? Was the effect of the PARPi concentration dependent?

The concentration of the PARPi were given in the methods section. Following recent publications (Zandarshvili et al. 2020) we used a high excess of PARPi (200 μ M) in order to maximize the effect of the inhibitors.

It seems that the analysis of PARPi effects on the DNA state distribution and/or interconversion rates is consistent with the classification of PARPi based on allosteric effects on DNA binding and hydrogen exchange, but it is not clear how separated they are. In some cases, the smFRET efficiency histograms show fairly clear evidence of a shift in populations (EB-47 and niraparib), and the interconversion times in Table 1 are also fairly distinct (although it would be useful to have some statistical analysis of the differences). For the other cases (talazoparib, veliparib, rucaparib, olaparib), the smFRET efficiency histograms do not appear to show major differences, but the interconversion times are distinct (statistical analysis of the differences would also help here). Is there a difference between niraparib/rucaparib versus veliparib that can be inferred from these differences?

By the data shown in this manuscript we have found that PARP-1 bound DNA is dynamically switching between the previously identified kinked state, and a linear state which had not been described in the context of DNA damage before. The resulting smFRET histograms show a skewed conformation rather than two clearly resolved peaks (compare Supplementary Figure S2). The histograms were fitted with three gaussians (sometimes with comparatively large widths) to account for heterogeneity. Thus rather than analyzing subtle changes in the shape of the distribution, we have performed a rigorous analysis of the underlying dynamics, i.e. the relaxation time τ_R , including statistical errors as described in detail in the Methods section. The data including error bars are presented in Table 1 of the manuscript. We find that for the class I inhibitor (EB-47) the relaxation time is shortened, indicative of a higher rate from the linear state back to the kinked state. Class II inhibitors show relaxation times on the order of that in absence of inhibitors (yet slightly lower) indicating a comparable rate of transfer from the linear state and thus also similar smFRET histograms. Class III inhibitors

show an increase in relaxation time which can be interpreted as a reduced rate of transfer from the linear state to the kinked state and thus show a more pronounced low-FRET population. Between the class II inhibitors olaparib and talazoparib the relaxation times are virtually identical, while for class III inhibitors rucaparib has the longest relaxation time. We have expanded the discussion of the results in the paper, to clarify the results.

"In this structure, the gapped DNA adopts a highly kinked conformation, similar to what had been seen for nicked DNA in early electron microscopy studies (Le Cam et al. 1994), F2 is bound to the side of the gap bearing the 3' terminus, F1 binds to the side bearing the 5' terminus, and the two zinc fingers form interdomain contacts with one another."
I suggest breaking this sentence in two, by placing a period after the Le Cam reference.

We have made the suggested changes to the manuscript.

"which might help to explain the different reported potencies of these inhibitors in vitro and in vivo (Rudolph, Jung, and Luger 2022)."
It seems appropriate to cite the primary literature reporting these differences.

We have added these references to the manuscript.

The below reference was listed twice (and has a formatting issue).
Langelier, Marie-france, Poly Adp-ribosyl, Jamie L Planck, Swati Roy, and John M Pascal. 2012. "Structural Basis for DNA Damage-Dependent Poly(ADP-Ribosyl)ation by Human PARP-1" 336 (6082): 728-32. 1032 <https://doi.org/10.1126/science.1216338>.

We have corrected this error.

The publication PMID: 21233213 really should be referenced in this study, as structures from this publication (3oda, 3odc) are specifically cited in the methods.

We have now cited this reference.

"Yet, we find that the diffusion time determined by FCS shows an increase in presence of XRCC1 (Supp. Fig. S12a lower panel)"
Seems that this should be Fig. S12c.

We have corrected this error.

"Interestingly, class III PARP-1 inhibitors such as niraparib have the opposite effect, the kinked state is de-stabilised as predicted by the hydrogen exchange data (Zandarashvili et al. 2020) and the linear conformation becomes more prominent (Fig. 6b)."
Shouldn't this be Figure 6c?

We have corrected this error.

REVIEWERS' COMMENTS

Reviewer #3 (Remarks to the Author):

The authors have addressed the remaining concerns with the revised manuscript.